# DiRong1.0: A distributed implementation for improving routing network generation in model coupling

Hao Yu[1], Li Liu[1,3], Chao Sun[1], Ruizhe Li[1], Xinzhu Yu[1], Cheng Zhang[1], Zhiyuan Zhang[2], Bin Wang[1,4]

[1] Ministry of Education Key Laboratory for Earth System Modeling, Department of Earth System Science, Tsinghua University, Beijing, China

[2] Hydro-Meteorological Center of Navy China, Beijing China

[3] Southern Marine Science and Engineering Guangdong Laboratory (Zhuhai), China

[4] State Key Laboratory of Numerical Modeling for Atmospheric Sciences and Geophysical Fluid Dynamics (LASG), Institute of Atmospheric Physics, Chinese Academy of Sciences, Beijing, China

*Correspondence to*: Li Liu (liuli-cess@tsinghua.edu.cn)

**Abstract.** A fundamental functionality of model coupling in an Earth system model is to efficiently handle data transfer between component models. An approach of *MxN* communication following a routing network has been used widely used for data transfer, and routing network generation becomes a major step required to initialize data transfer functionality. Some existing coupling software such as the Model Coupling Toolkit (MCT) and the existing versions of the Community Coupler

(C-Coupler) employ a global implementation of routing network generation that relies on gather/broadcast communications, which can be very inefficient under a case of a large number of processes. This is an important reason why the initialization cost of a coupler increases with the number of processor cores. In this paper, we propose a **D**istributed **i**mplementation for **Ro**uting **n**etwork **g**eneration, version 1.0 (DiRong1.0), which does not introduce any gather/broadcast communication. Empirical evaluations show that DiRong1.0 is much more efficient than the global implementation. DiRong1.0 has already

been implemented in C-Coupler2, and we believe that some other couplers can also benefit from it.

## 1 Introduction

Coupled Earth system models and numerical weather forecasting models highly depend on existing couplers (Hill et al., 2004; Craig et al., 2005; Larson et al., 2005; Balaji et al., 2006; Redler et al., 2010; Craig et al., 2012; Valcke, 2013; Liu et al., 2014;

Hanke et al., 2016; Craig et al., 2017; Liu et al., 2018). A coupler combines different component models into a whole system,
and handles data interpolation between different model grids and data transfer between component models (Valcke, 2012).

The process of data interpolation generally requires two major steps: preparing remapping weights, that are read from a file or are calculated online when initializing the coupler, and conducting parallel interpolation calculations based on sparse matrix-vector multiplication with the remapping weights throughout the coupled model integration. Couplers perform data transfer
by transferring scalar variables or fields on a model grid (hereafter called gridded fields) from one component model to another via Message Passing Interface (MPI). Component models are often parallelized by decomposing the cells of a model grid into distinct subsets, each of which is assigned to an MPI process for cooperative concurrent computation (e.g., the sample parallel decompositions in Fig. 1a and 1b). To efficiently transfer gridded fields in parallel, Jacob et al. (2005) proposed an approach of *MxN* communication (called the *MxN* approach) following a routing network, where each pair of processes from two
component models have a communication connection only when they share a common grid cell (for example, Fig. 1c). The *MxN* approach has been used in existing couplers for more than ten years. As the parallel decompositions of component models generally remain constant throughout the whole integration, a routing network can also remain constant. Thus, the *MxN* approach is realized through two major steps: generating the routing network when initializing the coupler and transferring gridded fields based on the routing network throughout the coupled model integration.


Due to the trend in model development towards higher grid resolutions and the resulting increased computation, the parallel efficiency of a coupled model on modern high-performance computers has become more critical. Any module in a coupled model, including the coupler, can impact the parallel efficiency of the whole model. Most existing couplers achieve scalable data transfer and data interpolation throughout the coupled model integration, i.e., the data transfer and data interpolation are
generally faster when using more processor cores. However, experiences with OASIS3-MCT and C-Coupler2 have shown that the initialization cost of a coupler can increase rapidly when using more processor cores (Craig et al., 2017; Liu et al., 2018). A further investigation based on MCT shows that the initialization of data transfer (i.e., generating routing networks) is an important source of the initialization cost (see Fig. 2).

This paper explores the first step toward lowering the initialization cost of a coupler by focusing on the generation of routing networks, and proposes a new **D**istributed **i**mplementation for **Ro**uting **n**etwork **g**eneration, version 1.0 (DiRong1.0). The evaluation based on C-Coupler2 shows that it is much faster than the existing approach. The remainder of this paper is organized as follows. We investigate the existing implementations of routing network generation in Section 2, present and then evaluate DiRong1.0 in Sections 3 and 4, respectively, and conclude with a discussion of this work in Section 5.

## 2 Existing implementations of routing network generation

In some existing coupling software such as MCT and C-Coupler, the global information of a parallel decomposition is originally distributed among all processes of a component model. This is because a process only records its local parallel decomposition on the grid cells assigned to it. Therefore, these couplers generally use the following four steps for generating a routing network between the parallel decompositions of a source (*src*) and a destination (*dst*) component model.

1) Gathering global parallel decomposition: the *src*/*dst* root process gathers the global information of the *src*/*dst* parallel decomposition from all *src*/*dst* processes.

2) Exchanging global parallel decomposition: the *src*/*dst* root process first exchanges the *src*/*dst* global parallel decomposition with the *dst*/*src* root process, and then broadcasts the *dst*/*src* global parallel decomposition to all *src*/*dst* processes.

3) Detecting common grid cells: each *src*/*dst* process detects its common grid cells with each *dst*/*src* process based on its local parallel decomposition and the *dst*/*src* global parallel decomposition.

4) Generating the routing network: each *src*/*dst* process generates its local routing network according to the information about common grid cells.

Assuming that each of the *src* and *dst* component models uses $K$ processes on a grid of size $N$ (i.e., the grid has $N$ cells), the first and second steps when using C-Coupler correspond to gather/broadcast communications with a time complexity of at least $O(N*logK)$ and a memory complexity of $O(N)$. The average time complexity of the third step is $O(N)$, as C-Coupler first generates a map corresponding to the global parallel decomposition and then detects common cells by looking at the map. Although this implementation tries to lower the time complexity, it introduces inefficient and irregular memory access. As the last step does not depend on any global parallel decomposition, its average time complexity is $O(N/K)$. MCT (as well as CPL6/CPL7 and OASIS3-MCT, which employ MCT for data transfer) has similar complexities to C-Coupler, even if a compressed global index description is, in the case of regular parallel decompositions, used to reduce the memory and the time required to detect common grid cells corresponding to regular parallel decompositions (the compressed description may not work for irregular, such as round-robin, parallel decompositions).

Given the gather/broadcast communications and the corresponding time complexity of $O(N*logK)$, and the time complexity of $O(N)$ corresponding to common grid cell detection, such existing implementations of routing network generation are of course inefficient with an increasing number of processor cores. Moreover, due to the memory complexity of $O(N)$, more memory is consumed as the model grid becomes finer.

In the following, the existing implementations relying on gather/broadcast communications will be called "global routing network generation".

## 3 Design and implementation

### 3.1 Overall design

The design and implementation of DiRong1.0 significantly benefits from the general idea of distributed directories (Pinar and Hendrickson, 2001), which have already been used in coupler development (Theurich et al., 2008; Hanke et al., 2016). Another different kind of specific distributed directories is defined and used in DiRong1.0.

Each cell of a grid can be numbered with a unique index from 1 to $N$ called the "global" cell index, while each grid cell assigned to the same process can be numbered with a unique "local" cell index. Thus, the information of a given parallel decomposition can be recorded as a Cell Local–Global Mapping Table (CLGMT), each element of which is a triple of global cell index, process ID, and local cell index. For example, Tables 1 and 2 are the CLGMTs corresponding to the parallel decompositions in Fig. 1a and 1b, respectively.

Generally, the CLGMT entries of a parallel decomposition are distributed among the processes of a component model, which means a process only stores part of the CLGMT. The distribution of the CLGMT entries is determined by the model but not the coupler. The key idea of existing global implementations is to reconstruct the global CLGMT of the peer parallel decomposition in each process for routing network generation. To be an efficient solution though, DiRong1.0 should be fully based on a distributed CLGMT without reconstructing any global CLGMT.

Motivated by the above analysis, the key challenge in DiRong1.0 is achieving efficient rearrangement of the original distribution of the CLGMT entries of a given parallel decomposition into a regular intermediate distribution, and efficiently generating the routing network based on the intermediate distribution. Specifically, we employ a regular intermediate distribution that evenly distributes the CLGMT entries among processes based on the global cell indices placed in ascending order. Such an intermediate distribution is not only simple, but it also enables a straightforward rearrangement of the CLGMT entries into the intermediate distribution via a sorting procedure similar to distributed sort. With that, DiRong1.0 takes the following major steps to generate a routing network between the *src* and *dst* component models.

1) The *src/dst* component model rearranges the original distribution of the CLGMT entries of the *src/dst* parallel decomposition into the regular intermediate distribution.

2) The *src* and *dst* component models exchange the CLGMT entries in the intermediate distributions.

3) Based on the *src* and *dst* CLGMT entries in the intermediate distributions, each *src/dst* process generates table entries of

the sharing relationship, which describes how each grid cell is shared between the processes of the *src* and *dst* component models.

     4)    The *src/dst* component model rearranges the intermediate distribution of the entries in the sharing relationship table (SRT) into the original distribution of the CLGMT entries of the *src/dst* parallel decomposition.

     5)    Each *src/dst* process generates its local routing network based on the local SRT entries.


The remainder of this section details the implementation of each major step, except the last one because it is similar to the last major step in the global implementation.

## 3.2 Rearranging CLGMT entries within a component model

The rearrangement of CLGMT entries within a component model is achieved via a divide-and-conquer sorting procedure, similar to a merge sort using the global cell index as the keyword. This procedure first sorts the CLGMT entries locally in each process, and then iteratively conducts a distributed sort via a main loop of *logK* iterations, where *K* is the number of processes of the *src/dst* component model. In each iteration, processes are divided into distinct pairs and the two processes in each pair swap the CLGMT entries based on a point-to-point communication. Figure 3 shows an example of the distributed sort

corresponding to the CLGMT entries in Table 1, and Table 3 shows the distributed CLGMT after rearranging the CLGMT entries in Table 2. As shown in Fig. 3, the distributed sort employed in DiRong1.0 uses a similar butterfly communication pattern to the optimized MPI implementations of various collective communication operations (Brooks, 1986; Thakur et al., 2005).

## 145   3.3 Exchanging CLGMT entries between component models

After the rearrangement of the CLGMT in a component model, the CLGMT entries are sorted into ascending order based on their global cell index and are evenly distributed among processes. The CLGMT entries reserved in each process therefore have a determinate and non-overlapping range of global cell indices, and such a range can be easily calculated from the grid size, the total number of processes, and the process ID. Thus, it is straightforward to calculate the overlapping relationship of

the global cell index range between a *src* process and a *dst* process. As it is only necessary to exchange CLGMT entries between a pair of *src* and *dst* processes with overlapping ranges, point-to-point communications suffice to handle the exchange of the CLGMT entries.

### 3.4 Generation of SRT

Following the previous major step, each process reserves two sequences of CLGMT entries corresponding to the *src* and *dst* parallel decompositions. Given that the two sequences contain *n1* and *n2* entries, respectively, the time complexity of detecting the sharing relationship is $O(n1+n2)$, because the entries in each sequence have already been ordered by ascending global cell index, and a procedure similar to the kernel of merge sort, which merges two ordered data sequences, can handle such a detection.


To record the sharing relationship, an SRT entry is designed as a quintuple of global cell index, *src* process ID, *src* local cell index, *dst* process ID, and *dst* local cell index. Given a quintuple $<q_1,q_2,q_3,q_4,q_5>$, the data on global cell $q_1$ in the *src* component model, corresponding to local cell $q_3$ in process $q_2$, is transferred to local cell $q_5$ in process $q_4$ in the *dst* component model. Table 4 shows the SRT in the *src* component model calculated from the rearranged, distributed CLGMT entries in Fig. 3 and
Table 3.

It is possible that multiple *src* CLGMT entries correspond to the same global cell index. In such a case, any *src* CLGMT entry can be used for generating the corresponding SRT entries, because the *src* component model guarantees that the data copies on the same grid cell are identical. Given a *dst* CLGMT entry, if there is no *src* CLGMT entry with the same global cell index,
no SRT entry will be generated. In the case that multiple *dst* CLGMT entries correspond to the same global cell index and there is at least one *src* CLGMT entry with the same global cell index, a SRT entry will be generated for every *dst* CLGMT entry.

### 3.5 Rearranging SRT entries within a component model

After the previous major step, the SRT entries are distributed among processes of a component model according to the intermediate distribution. Because a process can only use the SRT entries corresponding to its local cells for the last major step of local routing network generation, the SRT entries need to be rearranged among the processes of a component model. We find that such a rearrangement can be achieved via a sorting procedure similar to a distributed sort using the *src*/*dst* process ID as a keyword, or even via the sorting procedure implemented in the first major step. Tables 5 and 6 show the SRT entries
distributed in the *src* and *dst* component model, respectively, after the rearrangement.

### 3.6 Time complexity and memory complexity

As DiRong1.0 does not reconstruct the global CLGMT, it does not rely on any gather/broadcast communication and its average memory complexity is $O(N/K)$ for each process. Because the implementation of its most time-consuming steps are similar to

a merge sort, and the time complexity of a merge sort is $O(N*logN)$, the average time complexity of DiRong1.0 for each process is $O(N*(logN)/K)$, and the average communication complexity is $O(N*(logK)/K)$.

To facilitate the implementation of the sorting procedure, we force the number of processes in the first to fourth major steps to be the maximum power of 2 ($2^n$) no larger than the total number of processes of the *src/dst* component model. For a process whose ID $I$ is not smaller than $2^n$, its CLGMT entries are merged into the process with ID $I-2^n$ before the first major step, and the SRT entries corresponding to it are obtained from the process with ID $I-2^n$ after the fourth major step. This strategy does not change the aforementioned time complexity and memory complexity of DiRong1.0, as $2^n$ is larger than half of the total number of processes.

## 4 Evaluation

To evaluate DiRong1.0, we implement it in C-Coupler2, which enables us to compare it with the original global routing network generation. We develop a toy coupled model for the evaluation consisting of two toy component models and C-Coupler2, which allows us to flexibly change the model settings in terms of grid size and number of processor cores (processes). The toy coupled model is run on a supercomputer, where each computing node includes two Intel Xeon E5-2678 v3 CPUs (Intel(R) Xeon(R) CPU, 24 processor cores in total), and all computing nodes are connected with an InfiniBand network. The codes are compiled by an Intel Fortran and C++ compiler at the optimization level O2 using an Intel MPI library (2018 Update 2). A maximum of 6400 cores are used for running the toy coupled model, and all test results are from the average of multiple runs.

In Table 7 to Table 10, we evaluate the effect of varying the number of processes; the two component models use the same number of processor cores. For a grid size of 500,000 (Table 7), the execution time of DiRong1.0 does not significantly decrease when using more processor cores. This result is reasonable, although it does not match the time complexity of DiRong1.0. The communication complexity of DiRong1.0 is O($N*(logK)/K$), where $logK$ stands for the number of point-to-point communications in each process and $N/K$ stands for the average message size in each communication. The average message size corresponding to Table 7 is small (about 160 KB with 60 cores and about 6 KB with 1600 cores for each toy component model), but the execution time of point-to-point communication does not vary linearly with message size and may be unstable when the message size is small. In contrast to DiRong1.0, the execution time of the global implementation increases rapidly with increasing number of cores. As a result, DiRong1.0 outperforms the global implementation more significantly when using more cores. When the grid size increases (e.g., from 4,000,000 in Table 8 to 32,000,000 in Table 10), DiRong1.0 still significantly outperforms the global implementation and also has better scalability.

Considering that a model can use more processor cores for acceleration when its resolution becomes finer, we further evaluate the weak scalability of DiRong1.0 by concurrently increasing the grid size and number of cores to achieve similar numbers of grid points per process. As shown in Table 11, the execution time of DiRong1.0 increases slowly, whereas the execution time of the global implementation increases rapidly with larger grid sizes and increasing number of cores. This demonstrates that DiRong1.0 achieves much better weak scalability than the global implementation.

## 5 Conclusion and discussion

This paper proposes a new distributed implementation, DiRong1.0, for routing network generation. It is much more efficient than the global implementation as it does not introduce any gather/broadcast communication and it achieves much lower complexity in terms of time, memory, and communication. This conclusion is demonstrated by our evaluation results. DiRong1.0 has already been implemented in C-Coupler2. Its code is publicly available in a C-Coupler2 version and will be further used in future C-Coupler versions. We believe that some existing couplers such as MCT, OASIS3-MCT, and CPL6/CPL7 can also benefit from DiRong1.0, as it accelerates the routing network generation as well as the coupler initialization.

We did not evaluate the impact of DiRong1.0 on the total time of a model simulation, because this impact can be relative. The overhead of routing network generation as well as coupler initialization is trivial for a long simulation (e.g., hundreds of model days or even hundreds of model years), but may be significant for a short simulation (e.g., several model days or even several model hours in weather forecasting (Palmer et al., 2008; Hoskins, 2013)). Data assimilation for weather forecasting may require a model to run for only several model hours or even less time. In an operational model, there is generally a time limitation on producing forecasting results (for example, finishing a five-day forecast in two hours), and thus developers always have to carefully optimize various software modules, especially when the model resolution becomes finer. In fact, one of the primary motivations for the development of DiRong1.0 was to accelerate the initialization of C-Coupler2 for an operational coupled model used in China.

Another main reason for developing DiRong1.0 is that routing network generation will become more important in later versions of C-Coupler. Recently, a new framework was developed for weakly coupled **e**nsemble **d**ata **a**ssimilation (EDA) based on C-Coupler2, named DAFCC1 (Sun et al., 2020). DAFCC1 will be an important part of C-Coupler3, the next version of C-Coupler. For users wanting the atmosphere component of a coupled system to perform EDA, DAFCC1 will automatically generate an ensemble component corresponding to all ensemble members of the atmosphere component for calling the DA algorithm, and will automatically conduct routing network generation for the data transfers between the ensemble component and each ensemble member. Thus, routing network generation will be more frequently used in EDA with DAFCC1. For example, given 50 ensemble members, the routing network generation with the ensemble component will be conducted at least 50 times.

250

We note that the current sequential read of a remapping weight file is another drawback of C-Coupler2. Similar to Hanke et al. (2016), we will design a specific distributed directory for reading in the remapping weights in parallel, which will allow the remapping weights to be efficiently redistributed among processes based on DiRong1.0. Currently, C-Coupler2 employs a simple global representation for horizontal grids, which means that each process retains all points of a horizontal grid in memory. The global representation will become a bottleneck in at least two aspects. First, it will consume too much memory to run a model simulation. For example, given a horizontal grid with 16,000,000 points, the memory required to keep it in each process is large: about 1.3 GB, provided that each point has four vertices and the data type is double precision. Second, the initialization of the data interpolation functionality requires model grids to be exchanged between different component models, which introduces global communications (e.g., broadcast) for the global grid representations. To address this bottleneck, we will design and develop a distributed grid representation that can be viewed as a specific distributed directory, and will enable an efficient redistribution of horizontal grid points among processes based on DiRong1.0.

*Code availability.* The source code of DiRong1.0 can be viewed and run with C-Coupler2 and the toy coupled model via https://doi.org/10.5281/zenodo.3971829. The MCT version corresponding to Fig. 2 is 2.10 (https://www.mcs.anl.gov/research/projects/mct/)

*Author contributions.* HY was responsible for code development, software testing, and experimental evaluation of DiRong1.0, and co-led paper writing. LL initiated this research, was responsible for the motivation and design of DiRong1.0, supervised HY, and co-led paper writing. CS, RL, XY, and CZ contributed to code development and software testing. ZZ and BW contributed to the motivation and software testing. All authors contributed to the improvement of ideas and paper writing.

*Competing interests.* The authors declare that they have no conflict of interest.

*Acknowledgements*. This work was supported in part by the Natural Science Foundation of China (grant no. 42075157), and was jointly supported in part by the National Key Research Project of China (grant no. 2017YFC1501903).

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

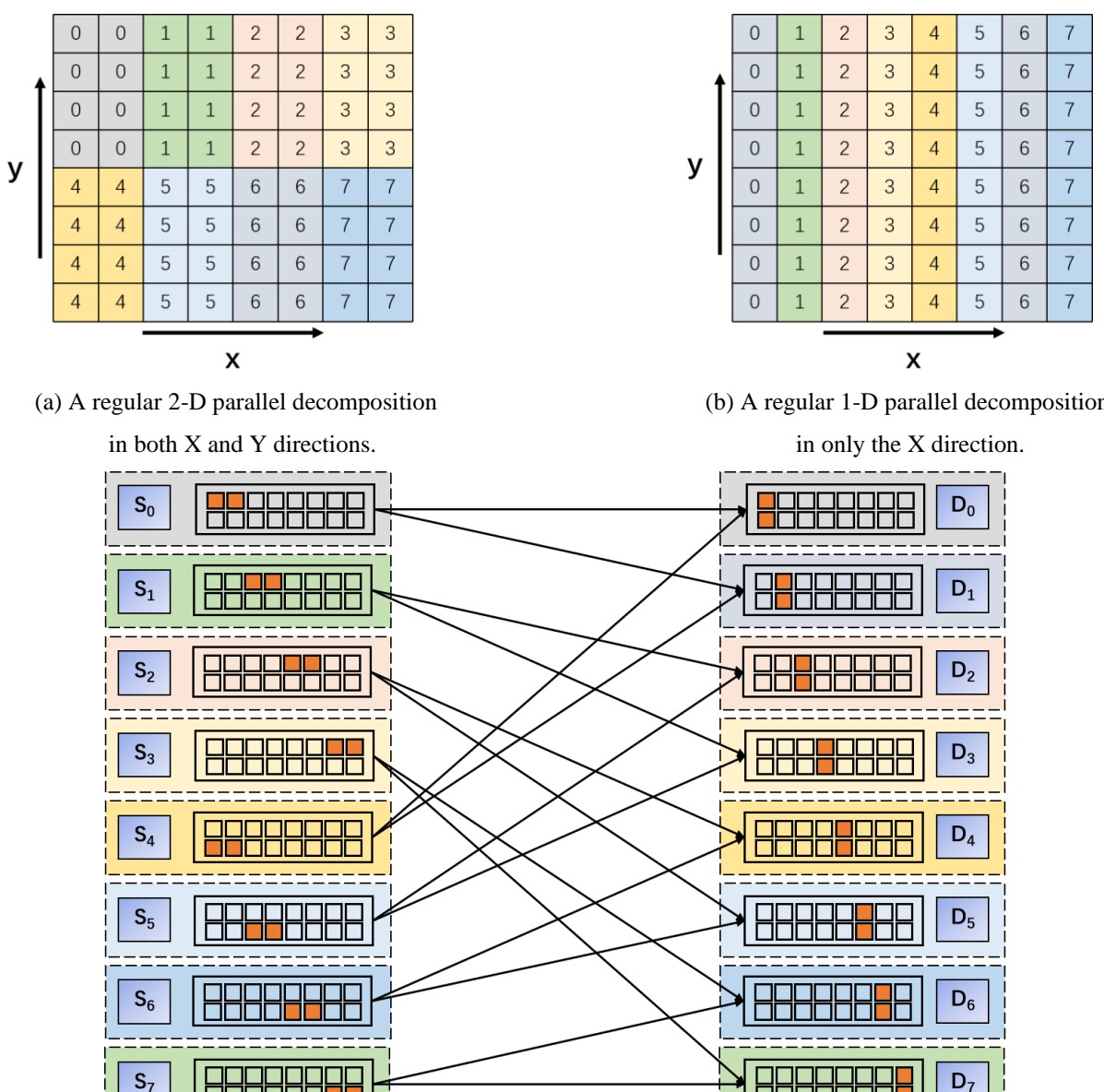

(a) A regular 2-D parallel decomposition in both X and Y directions.

(b) A regular 1-D parallel decomposition in only the X direction.

(c) The routing network from the parallel decomposition in Fig. 1a (**S**ource) to the parallel decomposition in Fig. 1(b) (**D**estination).

**Figure 1. Two sample parallel decompositions of an 8 × 8 grid under eight processes (Fig. 1a and 1b) and the routing network between them (Fig. 1c). Each color corresponds to a process. The left (for processes S₀~S₇) and the right (for processes D₀~D₇) in Fig. 1c correspond to the parallel decompositions in Fig. 1a and 1b respectively. A smallest square in Fig. 1c represents four continuous grid points in a column in Fig. 1a or 1b, where a square that corresponds to local grid points of a process is in brown.**

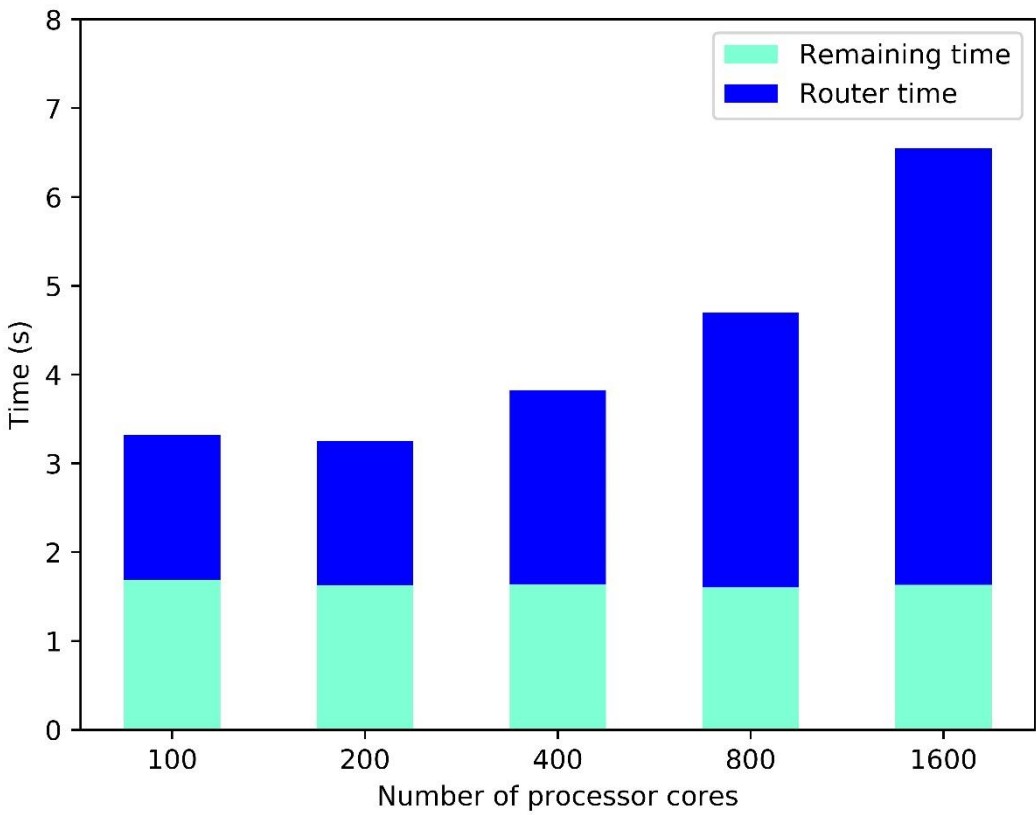


**Figure 2. The total time of routing network generation (router time) and the remaining time for initializing a two-way MCT coupling between two toy component models. One toy component model uses a longitude–latitude grid with 4 million points and a regular 2-D parallel decomposition, while the other uses a cubed-sphere grid with a resolution of 0.3 degrees and a round-robin parallel decomposition. The time for reading an offline remapping weight file has been**
**taken into account in the remaining time. The supercomputer as well as the corresponding software stacks described in Section 4 are used for this test.**

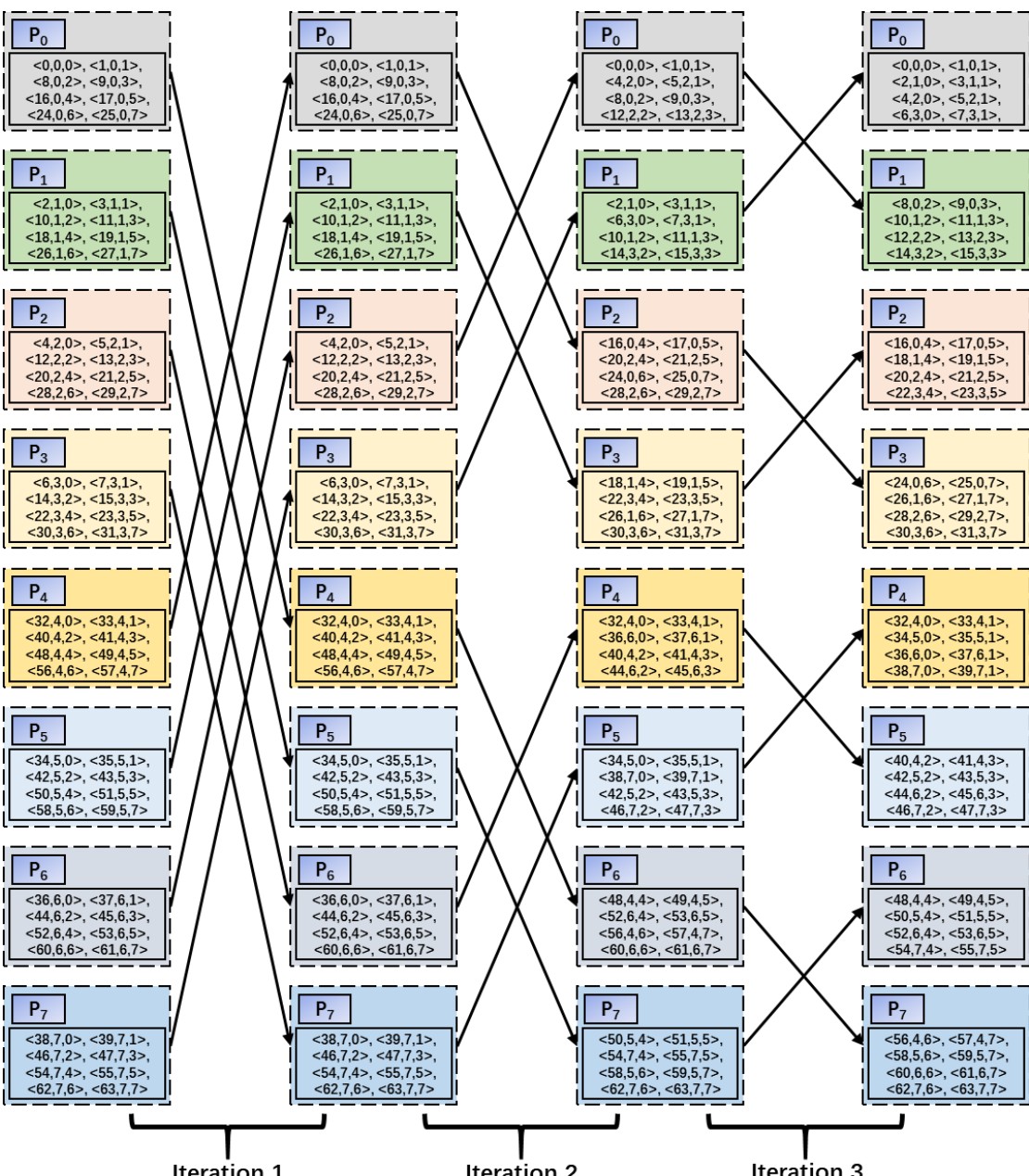

**Figure 3. The distributed sort corresponding to the CLGMT entries in Table 1. Each iteration makes the CLGMT entries with larger global cell indices reserved in the processes with larger IDs. For example, after the first iteration, the CLGMT entries with global cell indices between 0 and 31 are reserved in P0–P3, while the remaining CLGMT entries are reserved in P4–P7.**

**Table 1. The Cell Local–Global Mapping Table (CLGMT) of the parallel decomposition in Fig. 1a**

| Process ID | Cell Local–Global Mapping Table entries |
|---|---|
| 0 | <0,0,0>, <1,0,1>, <8,0,2>, <9,0,3>, <16,0,4>, <17,0,5>, <24,0,6>, <25,0,7> |
| 1 | <2,1,0>, <3,1,1>, <10,1,2>, <11,1,3>, <18,1,4>, <19,1,5>, <26,1,6>, <27,1,7> |
| 2 | <4,2,0>, <5,2,1>, <12,2,2>, <13,2,3>, <20,2,4>, <21,2,5>, <28,2,6>, <29,2,7> |
| 3 | <6,3,0>, <7,3,1>, <14,3,2>, <15,3,3>, <22,3,4>, <23,3,5>, <30,3,6>, <31,3,7> |
| 4 | <32,4,0>, <33,4,1>, <40,4,2>, <41,4,3>, <48,4,4>, <49,4,5>, <56,4,6>, <57,4,7> |
| 5 | <34,5,0>, <35,5,1>, <42,5,2>, <43,5,3>, <50,5,4>, <51,5,5>, <58,5,6>, <59,5,7> |
| 6 | <36,6,0>, <37,6,1>, <44,6,2>, <45,6,3>, <52,6,4>, <53,6,5>, <60,6,6>, <61,6,7> |
| 7 | <38,7,0>, <39,7,1>, <46,7,2>, <47,7,3>, <54,7,4>, <55,7,5>, <62,7,6>, <63,7,7> |

**Table 2. The Cell Local–Global Mapping Table (CLGMT) of the parallel decomposition in Fig. 1b**

| Process ID | Cell Local–Global Mapping Table entries |
|---|---|
| 0 | <0,0,0>, <8,0,1>, <16,0,2>, <24,0,3>, <32,0,4>, <40,0,5>, <48,0,6>, <56,0,7> |
| 1 | <1,1,0>, <9,1,1>, <17,1,2>, <25,1,3>, <33,1,4>, <41,1,5>, <49,1,6>, <57,1,7> |
| 2 | <2,2,0>, <10,2,1>, <18,2,2>, <26,2,3>, <34,2,4>, <42,2,5>, <50,2,6>, <58,2,7> |
| 3 | <3,3,0>, <11,3,1>, <19,3,2>, <27,3,3>, <35,3,4>, <43,3,5>, <51,3,6>, <59,3,7> |
| 4 | <4,4,0>, <12,4,1>, <20,4,2>, <28,4,3>, <36,4,4>, <44,4,5>, <52,4,6>, <60,4,7> |
| 5 | <5,5,0>, <13,5,1>, <21,5,2>, <29,5,3>, <37,5,4>, <45,5,5>, <53,5,6>, <61,5,7> |
| 6 | <6,6,0>, <14,6,1>, <22,6,2>, <30,6,3>, <38,6,4>, <46,6,5>, <54,6,6>, <62,6,7> |
| 7 | <7,7,0>, <15,7,1>, <23,7,2>, <31,7,3>, <39,7,4>, <47,7,5>, <55,7,6>, <63,7,7> |

**Table 3. The distributed CLGMT after rearranging the CLGMT entries in Table 2**

| Process ID | CLGMT entries |
|---|---|
| 0 | <0,0,0>, <1,1,0>, <2,2,0>, <3,3,0>, <4,4,0>, <5,5,0>, <6,6,0>, <7,7,0> |
| 1 | <8,0,1>, <9,1,1>, <10,2,1>, <11,3,1>, <12,4,1>, <13,5,1>, <14,6,1>, <15,7,1> |
| 2 | <16,0,2>, <17,1,2>, <18,2,2>, <19,3,2>, <20,4,2>, <21,5,2>, <22,6,2>, <23,7,2> |
| 3 | <24,0,3>, <25,1,3>, <26,2,3>, <27,3,3>, <28,4,3>, <29,5,3>, <30,6,3>, <31,7,3> |
| 4 | <32,0,4>, <33,1,4>, <34,2,4>, <35,3,4>, <36,4,4>, <37,5,4>, <38,6,4>, <39,7,4> |
| 5 | <40,0,5>, <41,1,5>, <42,2,5>, <43,3,5>, <44,4,5>, <45,5,5>, <46,6,5>, <47,7,5> |
| 6 | <48,0,6>, <49,1,6>, <50,2,6>, <51,3,6>, <52,4,6>, <53,5,6>, <54,6,6>, <55,7,6> |
| 7 | <56,0,7>, <57,1,7>, <58,2,7>, <59,3,7>, <60,4,7>, <61,5,7>, <62,6,7>, <63,7,7> |


**Table 4. The Sharing Relationship Table (SRT) calculated from the rearranged distributed CLGMT entries in Fig. 3 and Table 3**

| Process ID | Sharing Relationship Table entries |
|---|---|
| 0 | <0,0,0,0,0>, <1,0,1,1,0>, <2,1,0,2,0>, <3,1,1,3,0>, <4,2,0,4,0>, <5,2,1,5,0>, <6,3,0,6,0>, <7,3,1,7,0> |
| 1 | <8,0,2,0,1>, <9,0,3,1,1>, <10,1,2,2,1>, <11,1,3,3,1>, <12,2,2,4,1>, <13,2,3,5,1>, <14,3,2,6,1>, <15,3,3,7,1> |
| 2 | <16,0,4,0,2>, <17,0,5,1,2>, <18,1,4,2,2>, <19,1,5,3,2>, <20,2,4,4,2>, <21,2,5,5,2>, <22,3,4,6,2>, <23,3,5,7,2> |
| 3 | <24,0,6,0,3>, <25,0,7,1,3>, <26,1,6,2,3>, <27,1,7,3,3>, <28,2,6,4,3>, <29,2,7,5,3>, <30,3,6,6,3>, <31,3,7,7,3> |
| 4 | <32,4,0,0,4>, <33,4,1,1,4>, <34,5,0,2,4>, <35,5,1,3,4>, <36,6,0,4,4>, <37,6,1,5,4>, <38,7,0,6,4>, <39,7,1,7,4> |
| 5 | <40,4,2,0,5>, <41,4,3,1,5>, <42,5,2,2,5>, <43,5,3,3,5>, <44,6,2,4,5>, <45,6,3,5,5>, <46,7,2,6,5>, <47,7,3,7,5> |
| 6 | <48,4,4,0,6>, <49,4,5,1,6>, <50,5,4,2,6>, <51,5,5,3,6>, <52,6,4,4,6>, <53,6,5,5,6>, <54,7,4,6,6>, <55,7,5,7,6> |
| 7 | <56,4,6,0,7>, <57,4,7,1,7>, <58,5,6,2,7>, <59,5,7,3,7>, <60,6,6,4,7>, <61,6,7,5,7>, <62,7,6,6,7>, <63,7,7,7,7> |

**Table 5. The SRT entries distributed in the *src* component model after rearranging the SRT in Table 4**

| Process ID | Sharing Relationship Table entries |
|---|---|
| 0 | <0,0,0,0,0>, <1,0,1,1,0>, <8,0,2,0,1>, <9,0,3,1,1>, <16,0,4,0,2>, <17,0,5,1,2>, <24,0,6,0,3>, <25,0,7,1,3> |
| 1 | <2,1,0,2,0>, <3,1,1,3,0>, <10,1,2,2,1>, <11,1,3,3,1>, <18,1,4,2,2>, <19,1,5,3,2>, <26,1,6,2,3>, <27,1,7,3,3> |
| 2 | <4,2,0,4,0>, <5,2,1,5,0>, <12,2,2,4,1>, <13,2,3,5,1>, <20,2,4,4,2>, <21,2,5,5,2>, <28,2,6,4,3>, <29,2,7,5,3> |
| 3 | <6,3,0,6,0>, <7,3,1,7,0>, <14,3,2,6,1>, <15,3,3,7,1>, <22,3,4,6,2>, <23,3,5,7,2>, <30,3,6,6,3>, <31,3,7,7,3> |
| 4 | <32,4,0,0,4>, <33,4,1,1,4>, <40,4,2,0,5>, <41,4,3,1,5>, <48,4,4,0,6>, <49,4,5,1,6>, <56,4,6,0,7>, <57,4,7,1,7> |
| 5 | <34,5,0,2,4>, <35,5,1,3,4>, <42,5,2,2,5>, <43,5,3,3,5>, <50,5,4,2,6>, <51,5,5,3,6>, <58,5,6,2,7>, <59,5,7,3,7> |
| 6 | <36,6,0,4,4>, <37,6,1,5,4>, <44,6,2,4,5>, <45,6,3,5,5>, <52,6,4,4,6>, <53,6,5,5,6>, <60,6,6,4,7>, <61,6,7,5,7> |
| 7 | <38,7,0,6,4>, <39,7,1,7,4>, <46,7,2,6,5>, <47,7,3,7,5>, <54,7,4,6,6>, <55,7,5,7,6>, <62,7,6,6,7>, <63,7,7,7,7> |


**Table 6. The SRT entries distributed in the *dst* component model after rearranging the SRT in Table 4**

| Process ID | Sharing Relationship Table entries |
|---|---|
| 0 | <0,0,0,0,0>, <8,0,2,0,1>, <16,0,4,0,2>, <24,0,6,0,3>, <32,4,0,0,4>, <40,4,2,0,5>, <48,4,4,0,6>, <56,4,6,0,7> |
| 1 | <1,0,1,1,0>, <9,0,3,1,1>, <17,0,5,1,2>, <25,0,7,1,3>, <33,4,1,1,4>, <41,4,3,1,5>, <49,4,5,1,6>, <57,4,7,1,7> |
| 2 | <2,1,0,2,0>, <10,1,2,2,1>, <18,1,4,2,2>, <26,1,6,2,3>, <34,5,0,2,4>, <42,5,2,2,5>, <50,5,4,2,6>, <58,5,6,2,7> |
| 3 | <3,1,1,3,0>, <11,1,3,3,1>, <19,1,5,3,2>, <27,1,7,3,3>, <35,5,1,3,4>, <43,5,3,3,5>, <51,5,5,3,6>, <59,5,7,3,7> |
| 4 | <4,2,0,4,0>, <12,2,2,4,1>, <20,2,4,4,2>, <28,2,6,4,3>, <36,6,0,4,4>, <44,6,2,4,5>, <52,6,4,4,6>, <60,6,6,4,7> |
| 5 | <5,2,1,5,0>, <13,2,3,5,1>, <21,2,5,5,2>, <29,2,7,5,3>, <37,6,1,5,4>, <45,6,3,5,5>, <53,6,5,5,6>, <61,6,7,5,7> |
| 6 | <6,3,0,6,0>, <14,3,2,6,1>, <22,3,4,6,2>, <30,3,6,6,3>, <38,7,0,6,4>, <46,7,2,6,5>, <54,7,4,6,6>, <62,7,6,6,7> |
| 7 | <7,3,1,7,0>, <15,3,3,7,1>, <23,3,5,7,2>, <31,3,7,7,3>, <39,7,1,7,4>, <47,7,3,7,5>, <55,7,5,7,6>, <63,7,7,7,7> |

**Table 7. Performance of DiRong1.0 and the comparison with the original global routing network generation (Global) using different numbers of cores numbers and the grid size of 500,000.**

| Core number of each toy component model | DiRong1.0 | | Global | | Global/DiRong1.0 |
|---|---|---|---|---|---|
| | Time (s) | Speedup | Time (s) | Speedup | |
| 60 | 0.031 | 1.000 | 0.129 | 1.000 | 4.110 |
| 120 | 0.040 | 0.774 | 0.278 | 0.462 | 6.888 |
| 240 | 0.047 | 0.671 | 0.243 | 0.530 | 5.205 |
| 480 | 0.029 | 1.076 | 0.478 | 0.269 | 16.461 |
| 960 | 0.033 | 0.943 | 1.169 | 0.110 | 35.224 |
| 1600 | 0.034 | 0.912 | 1.737 | 0.074 | 50.641 |
| 3200 | 0.036 | 0.862 | 2.573 | 0.050 | 70.900 |


**Table 8. Performance of DiRong1.0 and the comparison with the original global routing network generation (Global) using different numbers of cores numbers and the grid size of 4,000,000.**

| Core number of each toy component model | DiRong1.0 | | Global | | Global/DiRong1.0 |
|---|---|---|---|---|---|
| | Time (s) | Speedup | Time (s) | Speedup | |
| 60 | 0.161 | 1.000 | 0.863 | 1.000 | 5.349 |
| 120 | 0.117 | 1.375 | 0.517 | 1.668 | 4.409 |
| 240 | 0.081 | 1.990 | 0.437 | 1.974 | 5.391 |
| 480 | 0.060 | 2.669 | 0.649 | 1.329 | 10.737 |
| 960 | 0.051 | 3.184 | 1.308 | 0.660 | 25.811 |
| 1600 | 0.045 | 3.548 | 1.949 | 0.443 | 42.858 |
| 3200 | 0.039 | 4.098 | 2.623 | 0.329 | 66.598 |


**Table 9. Performance of DiRong1.0 and the comparison with the original global routing network generation (Global) using different numbers of cores numbers and the grid size of 16,000,000.**

| Core number of each toy component model | DiRong1.0 | | Global | | Global/DiRong1.0 |
|---|---|---|---|---|---|
| | Time (s) | Speedup | Time (s) | Speedup | |
| 60 | 0.702 | 1.000 | 3.437 | 1.000 | 4.899 |
| 120 | 0.447 | 1.571 | 2.351 | 1.462 | 5.263 |
| 240 | 0.276 | 2.547 | 2.363 | 1.455 | 8.575 |
| 480 | 0.169 | 4.163 | 2.529 | 1.359 | 15.006 |
| 960 | 0.109 | 6.429 | 3.135 | 1.097 | 28.721 |
| 1600 | 0.106 | 6.628 | 3.065 | 1.121 | 28.956 |
| 3200 | 0.098 | 7.133 | 3.242 | 1.060 | 32.960 |

**Table 10. Performance of DiRong1.0 and the comparison with the original global routing network generation (Global) using different numbers of cores numbers and the grid size of 32,000,000.**

| Core number of each toy component model | DiRong1.0 | | Global | | Global/DiRong1.0 |
|---|---|---|---|---|---|
| | Time (s) | Speedup | Time (s) | Speedup | |
| 60 | 1.438 | 1.000 | 6.878 | 1.000 | 4.782 |
| 120 | 0.960 | 1.499 | 4.206 | 1.635 | 4.383 |
| 240 | 0.554 | 2.597 | 4.739 | 1.451 | 8.557 |
| 480 | 0.340 | 4.234 | 5.083 | 1.353 | 14.964 |
| 960 | 0.199 | 7.222 | 6.098 | 1.128 | 30.616 |
| 1600 | 0.176 | 8.182 | 5.758 | 1.195 | 32.756 |
| 3200 | 0.165 | 8.704 | 5.500 | 1.251 | 33.286 |

**Table 11. Performance of DiRong1.0 and the comparison with the original global routing network generation (Global) when concurrently increasing the grid size and number of cores.**

| Core number of each toy component model | Grid size | Execution time (s) of DiRong1.0 | Execution time (s) of Global | Global/ DiRong1.0 |
|---|---|---|---|---|
| 250 | 500,000 | 0.032 | 0.262 | 8.19 |
| 450 | 1,000,000 | 0.034 | 0.492 | 14.47 |
| 900 | 2,000,000 | 0.041 | 1.158 | 28.24 |
| 1600 | 4,000,000 | 0.045 | 1.949 | 43.31 |
| 3200 | 8,000,000 | 0.063 | 2.850 | 45.24 |