# Peer review of "DiRong1.0: A distributed implementation for improving routing network generation in model coupling"

_Geoscientific Model Development, 2020_

## Referee Comment (RC1) · Moritz Hanke (Referee) · 24 Apr 2020

**Summary**

This manuscript addresses the issue of the generation of a routing table, which is used to handle the data redistribution between two sets of processes. In climate modelling this occurs for example in the transfer of data between two component models.

The authors of the manuscript assume that the majority of existing coupling solutions use an inefficient algorithm to generate these routing tables and they attribute the inefficiency mainly to the use of collective communication.

[Figure]

The majority of the manuscript concentrates on presenting an algorithm for the generation of routing tables and evaluation of its performance. Even though the presented algorithm itself is well suited for this task, it is not completely new and is already being used by different coupling solutions and communication libraries.

Overall, the manuscript is well structured and sufficiently well written. However, due to the lack of substantial contribution to modelling science, I would not recommend this manuscript for publication in the Geoscientific Model Development.

**General**

The presented algorithm is basically a rendezvous algorithm, which uses a distributed directory. This was first introduced in [1]. I assume that you did not know about this paper and therefore did not reference it. This algorithm works by distributing data among the processes in a globally known decomposition. This is called "regular intermediate distribution" in the manuscript. Using this intermediate decomposition, accessing data without knowing the original decomposition and without gathering all data on a single process is easily possible. The use of distributed directories in a similar context is mentioned in the following references [2], [3], and [4].

To generate the distributed directory, the authors propose to use a parallel sort, which uses the global indices of the cells as sorting keys. Such algorithms are commonly known and are no new inventions.

The remainder of the presented algorithm does not contain any significant scientific contributions. Therefore, I do not think that this manuscript contains a substantial contribution to modelling science.

**Specific comments**

The manuscript describes the router network generation based on two predefined decompositions from two component models as being a fundamental functionality of a coupler. However, this routing table can also be a by-product, in case both components have different grids and the coupler generates interpolation weights online.

*Line 30: "there is almost no evidence of scalable initialization of a coupler" [...] "(Craig et al., 2017; Liu et al., 2018)"*
Both papers mentioned in this context contain figures with initialisation cost measurements (see figure 2 and figure 8 respectively). Scaling behaviour in these figures is indeed sub-optimal. However, the cause for this scaling behaviour is not explicitly attributed by either paper to the router generation. Your manuscript indicates that in fact this is the case without providing evidence. By recreating figure 8 from the second paper with your new router generation implementation, you could have confirmed (at least for the C-Coupler) this.
In Hanke et al., 2016 (cited by the manuscript) figure 3 (b) shows good scalability of the overall coupler initialisation for up to 3072 processes per component.

*Line 56: "almost all existing couplers use the following 4 steps for generating a routing network"*
This is a very strong claim, which I would not support (see [2], [3], and [4]).

*Paragraph 3.2 and figure 2:*
This is the description of a basic parallel sorting algorithm. A shorter paragraph and a reference to a respective paper would have been enough.

*Line 167-168: "it only utilizes point-to-point communications and does not rely on any collective communication"*
The parallel sorting algorithm has a complexity of O(log(n)) and a similar communication pattern as the MPI implementations of various collective communication operations[5]. Therefore, you could argue that you actually did use collective communication, which you implemented using point-to-point communication.

*Paragraph 4:*
This paragraph does not describe how these measurements where generated. Did you do a single run, average of multiple runs or average over multiple executions of the

algorithm within a single run? It is possible, that especially the first execution of this algorithm produces for some MPI implementations a much higher run time than the following ones.

*Figure 3:*
I would have preferred to have absolute runtimes instead of speedups for the evaluation of the individual algorithms and speedups only for the direct comparison between the two.

*Tables 1 to 6:*
In my opinion, these tables add no significant value to the understanding of the algorithm.

*Table 7:*
I assume the 1600 cores mean, that you used two toy components with 1600 cores each. I could not find an explicit description of this.
The core counts are rather odd. You mentioned that your nodes have 24 processors each. Therefore, I would have assume, that the core counts in your tests are multiples of 24.

**Remarks**

*Intel MPI library (3.2.2)*
Why did you use such an old version?
In my experiments Intel MPI often performed very poorly. Maybe give the most recent OpenMPI a try.

You propose to use a sorting algorithm with complexity O(log(n)) to generate the distributed directory. In some tests, I have seen that some MPI implementations introduce significant delays with such communication pattern, when being used for the first time in a run.
Alternatively, each process could compute for all its local points the destination rank

in the distributed directory. Using alltoall, you can exchange the number of points that need to be sent, in order to get directly from the original to the intermediate decomposition. Afterwards, alltoallv can redistribute the data in a single communication call. Depending on the MPI implementation alltoall can also have a complexity of O(log(n)). However, very little data is exchanged and it can be highly optimised within the MPI. The communication matrix for the alltoallv is probably very sparse. Hence, it can be implemented by the user using point-to-point communication. In my tests, this approach delivers really good results.

The code provided for this manuscript already uses some parts of ESMF for time management. Maybe have a look at ESMCI_MeshRedist.C and Zoltan/dr_dd.c in the original ESMF repository, these should contain algorithms very similar to the one proposed in this manuscript.

1: A. Pinar and B. Hendrickson, Communication Support for Adaptive Computation in Proc. SIAM Conf. on Parallel Processing for Scientific Computing, 2001.
2: https://www.earthsystemcog.org/site_media/projects/esmf/pres_0812_board_gerhard.pdf
3: http://www.cs.sandia.gov/Zoltan/ug_html/ug_util_dd.html
4: Hanke, M., Redler, R., Holfeld, T., and Yastremsky, M.: YAC 1.2.0: new aspects for coupling software in Earth system modelling, Geosci. Model Dev., 9, 2755–2769, https://doi.org/10.5194/gmd-9-2755-2016, 2016
5: Thakur, R., Rabenseifner, R., Gropp, W. (2005). Optimization of Collective Communication Operations in MPICH. The International Journal of High Performance Computing Applications, 19(1), 49–66. https://doi.org/10.1177/1094342005051521

---

## Short Comment (SC1) · 25 Apr 2020

Dear Mr. Moritz Hanke,

Thanks a lot for your review. The incorrect statements in the manuscript you point out and the references and the introduction about YAC and ESMF in the review will help us to further improve this manuscript as well as our development of C-Coupler.

After carefully reading the review, we find that there are significant misunderstands regarding the contribution of this manuscript. We wish more discussions with you before revising the manuscript.

In the following, we'd like to pose discussions about some review points. Please show us if some points in the discussions are wrong. Thanks a lot.

1. Line 167-168: "it only utilizes point-to-point communications and does not rely on any collective communication". The parallel sorting algorithm has a complexity of O(log(n)) and a similar communication pattern as the MPI implementations of various collective communication operations[5]. Therefore, you could argue that you actually did use collective communication, which you implemented using point-to-point communication.

RESPONSE: The "collective communication" in Line 167-168 means gather and broadcast that have been used for routing network generation. Figure 1 and 2 in this reply shows an example of the collective communications of gather and broadcast based on a binary tree, respectively. Although the parallel sorting algorithm has the same number of tree levels with these collective communications, which is around log(K) (where K is number of processes), it is different from these collective communications at each tree level at least in two aspects. First, all processes work at each level in the parallel sorting algorithm (for example, Figure 2 in the manuscript), while a part of processes are idled at most levels of these collective communications. Second, the global data is distributed evenly among all processes at each level in the parallel sorting algorithm, and thus the message size corresponding to each point-to-point communication is O(N/K) (where N is the global size of the grid), while the message size generally doubles following the reverse tree of the gather communication, the whole global data is transferred at each point-to-point communication in the broadcast communication, and thus the message size corresponding to each point-to-point communication is O(N). So, the average complexity of the parallel sorting algorithm is O(log(K)*N/K), and is much lower than the average complexity of these collective communications, O(log(K)*N), which has been mentioned at Line 69 of this manuscript.

In response to the misunderstanding arising from the statement Line 167-168, we will correct it when revising the manuscript.

2. Alternatively, each process could compute for all its local points the destination rank. Using alltoall, you can exchange the number of points that need to be sent, in order to get directly from the original to the intermediate decomposition. Afterwards, alltoallv can redistribute the data in a single communication call. Depending on the MPI implementation alltoall can also have a complexity of O(log(n)). However, very little data is exchanged and it can be highly optimised within the MPI. The communication matrix for the alltoallv is probably very sparse. Hence, it can be implemented by the user using point-to-point communication. In my tests, this approach delivers really good results.

RESPONSE: It is true that each process can easily compute the destination rank for all its local points, and then can easily prepare the parameters of sendcnts as well as sdisls for using MPI alltoallv. However, each process cannot compute recvcnts from its local points without extra communications, while recvcnts is necessary for using MPI alltoallv. In my opinion, extra collective communications will be required for computing recvcnts. If we are wrong, could you please show us a solution without any collective communications? Thanks a lot.

Moreover, even though very little data is generally exchanged, the communication matrix is probably very sparse, and alltoallv can be highly optimised within the MPI, we avoid to use MPI alltoallv throughout the C-Coupler development that targets commonality and wide usage. This is because we have experiences that MPI alltoallv introduced deadlocks and its performance was unstable and depended on the parallel decompositions and MPI versions. To make C-Coupler as reliable and stable as possible especially for operational usage, such risks are not allowed for us. After viewing the code file communicator mpi.c in YAC1.5.4 (thank you a lot for making the code publicly available for viewing), it seems that YAC does not use MPI alltoallv currently. Could you show us why or do you have plan to use MPI alltoallv in future YAC versions? Thanks a lot.

3. The presented algorithm is basically a rendezvous algorithm, which uses a distributed directory. This was fi̧rst introduced in [1]. I assume that you did not know

about this paper and therefore did not reference it. This algorithm works by distributing data among the processes in a globally known decomposition. This is called "regular intermediate distribution" in the manuscript. Using this intermediate decomposition, accessing data without knowing the original decomposition and without gathering all data on a single process is easily possible. The use of distributed directories in a similar context is mentioned in the following references [2], [3], and [4].

RESPONSE: Thanks a lot for introducing the distributed directory and the related works, which will be referenced and discussed when revising the manuscript. We are glad to know that YAC and ESMF has already benefited from distributed directory. After viewing the paper of YAC1.2.0 and the corresponding code files of YAC1.5.4, we learn that YAC use a hash table as the distributed directory for developing a parallel input scheme to read in the weights. This manuscript should be a new usage of distributed directory in coupler development. We find that the routing network generation for data transfer functionality in YAC and ESMF has not benefit from distributed directory. Specifically, the global search in YAC that may implicitly include routing network generation relies on gather and broadcast, while the routing network generation in ESFM seems to rely on the global representation of parallel decompositions (could you please show us the details if this point is wrong. Thanks a lot). Moreover, the "regular intermediate distribution" used in this manuscript is a specific distributed directory different from what is used in YAC, which makes accessing data without knowing the original decomposition and without gathering all data on a single process easily achieved by distributed sort.

4. To generate the distributed directory, the authors propose to use a parallel sort, which uses the global indices of the cells as sorting keys. Such algorithms are commonly known and are no new inventions. The remainder of the presented algorithm does not contain any signi ficant scienti fic contributions. Therefore, I do not think that this manuscript contains a substantial contribution to modelling science.

RESPONSE: Regarding this manuscript that is a technical article about couplers, our

primary goal and first contribution should be coupler improvement rather than developing a new distributed sorting algorithm. Before the code development, we have tried several times to search an existing distributed sorting algorithm that matches our requirements, in order to reduce the work for developing the distributed algorithm for routing network generation. Although parallel sorting algorithms have been widely studied, we finally failed and we have to develop a new sort algorithm implementation. Considering some existing couplers can benefit from this manuscript, we detailed this distributed sorting algorithm. Since it has been stated that "Such algorithms are commonly known and are no new inventions" in this review point, could you please show us the detailed references of the existing algorithms that are the same with the algorithm in this manuscript? Thanks a lot.

We note that, model development in this world includes a lot of papers and contributions that use an existing parameterization scheme S from an existing model A into another existing model B. So, we do believe that the employment of distributed directory for the parallel scheme of reading in the weights in YAC makes a substantial contribution to coupler development. As the distributed algorithm in this manuscript improves routing network generation, we also believe that it makes a substantial contribution.

Moreover, along with finer and finer resolutions in model development, grid size becomes larger and larger, and thus global search used in YAC, C-Coupler, etc., will become a significant memory bottleneck. We think that contributions relevant to distributed directory for settling this bottleneck would be welcome.

5. The manuscript describes the router network generation based on two predefi̇ned decompositions from two component models as being a fundamental functionality of a coupler. However, this routing table can also be a by-product, in case both components have different grids and the coupler generates interpolation weights online.

RESPONSE: It has been stated in Valcke et al. (2012) that, "couplers used in the geophysical community typically carry out similar functions such as managing data

transfer between two or more components, interpolating the coupling data between different grids, and coordinating the execution of the constituent models". Following this statement, the abstract of this manuscript states that, "it is a fundamental functionality of a coupler for Earth system modeling to efficiently handle data transfer between component models. Routing network generation is a major step for initializing the data transfer functionality". In MCT, OASIS3-MCT and CPL6/CPL7 that employ MCT, and C-Coupler, data transfer and data interpolation are implemented as two standalone functionalities and there is an explicit major step of generating routing information, although OASIS3-MCT and C-Coupler can generate interpolation weights online. That's why we can believe that some existing couplers can benefit from this manuscript.

Data transfer and data interpolation can also be implemented in an inseparable procedure, where an extra data interpolation between the same grids could be added the model coupling corresponding to the same grid. Data transfer and data interpolation should be side-by-side functionalities, which means that data transfer as well as the routing table can be viewed as a by-product of data interpolation, while data interpolation can also be viewed as a by-product of data transfer. No matter what the view is, the importance of both data transfer and data interpolation should not be impacted by the view.

6. Line 30: "there is almost no evidence of scalable initialization of a coupler" [...] "(Craig et al., 2017; Liu et al., 2018)". Line 56: "almost all existing couplers use the following 4 steps for generating a routing network" RESPONSE: We will correct these statements when revising the manuscript.

7. Paragraph 4: This paragraph does not describe how these measurements where generated. Did you do a single run, average of multiple runs or average over multiple executions of the algorithm within a single run? It is possible, that especially the first execution of this algorithm produces for some MPI implementations a much higher run time than the following ones.

RESPONSE: We use average of multiple runs but not average of over multiple executions of the algorithm within a single run, which will be stated when revising the manuscript. Although we also know that the second execution of an MPI algorithm can be much faster than the first execution due to the "hot spots" or "memory", we have to use average of multiple runs, because the routine network generation for a data transfer is conducted only one time in a run under real cases.

8. Intel MPI library (3.2.2) Why did you use such an old version? In my experiments Intel MPI often performed very poorly. Maybe give the most recent OpenMPI a try.

RESPONSE: Thanks a lot for this concern. The correct MPI version used for the evaluations in the manuscript is Intel MPI library (2018 Update 2). We are sorry about the wrong introduction of "Intel MPI library (3.2.2)". We can use another MPI version for further evaluation when revising the manuscript, if required.

Wish your further comments.

Many thanks again,

Li

Valcke, S., Balaji, V., Craig, A., Deluca, C., Dunlap, R., Ford, R. W., Jacob, R., Larson, J., Okuinghttons, R., Riley, G., Vertenstein, M: Coupling technologies for Earth System Modelling, Geosci. Model Dev., 5, 1589-1596, 2012
* * *
[Figure]

[Figure]

Figure 1. An example of MPI_gather following the communication network of a binary tree. The total 16000 data values are distributed evenly among the 16 processes before gathering.

[Figure]

Figure 2. Similar to Fig.1 but for MPI_bcast

---

## Referee Comment (RC2) · Moritz Hanke (Referee) · 28 Apr 2020

Dear Mr. Li Liu,

I am happy to engage in a discussion about the manuscript and hope you do not mind if move the discussion of topics not directly linked to the manuscript (incl. comments specific to YAC) to a personal email.

**Regarding the meaning of "collective communication":**
The MPI Standard defines this as: "communication that involves a group or groups of processes" [1]. Your manuscript did not explicitly limit the meaning to the subgroup

of collective algorithms like gather or broadcast. If limited your definition of "collective communication" to these operations, then your sorting algorithm has indeed a better complexity. However, the common meaning of "collective communication" also includes operations like reduction, barrier or alltoall.

**On the sorting algorithm:**
I may have been a little bit hasty regarding my comments concerning the common knowledge about your sorting algorithm. After some research, I could not find any paper describing an algorithm similar to yours. However, I also looked a little bit more into your algorithm. In the manuscript it is described that in each iteration all data of the local table is exchanged with the respective communication partner. This also matches with the actual implementation, if I understood it correctly. In figure 2 however, only a one-directional data transfer is depicted, which is actually sufficient for the algorithm to work correctly. In addition, instead of sending all entries in the local table, you have enough information available to only send the data that is actually required by the receiving process. If you apply these two optimisations, the movement of the table entries between the processes is more or less identical to that of a collective alltoallv, if implemented using Brook's algorithm [2][3].

Therefore, I would interpret your sorting algorithm as a modified alltoallv. And by the definition of "collective communication" in the MPI Standard, your algorithm itself is a collective operation.

**Focus of the manuscript:**
The title, the abstract and the remaining contents of the manuscript indicate for me a major focus on the introduction of a new algorithm and its performance. In that case, I still stand by previous assessment, that this is not enough contribution to modelling science and I would still not recommend this manuscript for publication.

Yet, I agree with you on your work potentially being beneficial to other software used in modelling science. Your manuscript could focus more on C-Coupler and its perfor­mance. After a more detailed analysis of the coupler initialisation, which identifies the

router initialisation as a significant issue, you could provide your algorithm as a solution for this. This might require some more performance data detailing the impact of router initialisation on the overall initialisation of the coupler. And as mentioned in my review, I would prefer a different presentation of the data shown in figure 3. Since, the manuscript also explicitly mentions the algorithm within MCT, you could also add a comparison with its initialisation.

With best regards,
Moritz Hanke

1: MPI: A Message-Passing Interface Standard (Version 3.1) Message Passing Interface Forum (2015) http://www.mpi-forum.org/docs/
2: Brooks, E.D. The butterfly barrier. Int J Parallel Prog 15, 295–307 (1986). https://doi.org/10.1007/BF01407877
3: Thakur, Rajeev. (2005). Optimization of Collective Communication Operations in MPICH. International Journal of High Performance Computing Applications. 19. 49-66. 10.1177/1094342005051521.

---

## Short Comment (SC2) · 28 Apr 2020

Dear authors,

in my role as Executive editor of GMD, I would like to bring to your attention our Editorial version 1.2:

https://www.geosci-model-dev.net/12/2215/2019/

This highlights some requirements of papers published in GMD, which is also available on the GMD website in the 'Manuscript Types' section:

http://www.geoscientific-model-development.net/submission/manuscript_types.html

[Figure]

In particular, please note that for your paper, the following requirement has not been met in the Discussions paper:

- "The main paper must give the model name and version number (or other unique identifier) in the title."

Please add the name/acronym of DaRong and a version number in the title upon your revised submission to GMD. As DaRong is the new development and you named it anyway, it must be named in the title.

Yours,

Astrid Kerkweg

———————————————

---

## Short Comment (SC3) · 29 Apr 2020

Dear Mr. Moritz Hanke,

Thanks a lot for your new comments and suggestions, which will help us further improve the manuscript and the code implementation.

In the following, we'd like to give responses to your comments and suggestions.

1. Regarding the meaning of "collective communication".

RESPONSE: You are right that collective communication includes operations like reduction, barrier or alltoall, besides gather and broadcast. We should directly use broad-

cast for correction rather than collective communication. The manuscript will be modified accordingly.

2. On the sorting algorithm.

RESPONSE: We agree that our sorting algorithm can be viewed as an optimized implementation of collective alltoallv (using Brook's algorithm) specifically for routing network generation. We have improved the code for only sending the data that is actually required by the receiving process. The manuscript will be modified accordingly. Thanks a lot for your suggestion.

3. About the focus of the manuscript.

RESPONSE: We agree that the "new" "algorithm" throughout the manuscript does not match the contribution of this work. We will improve the title, abstract and some contents (including Figure 3) when revising the manuscript.

Thanks a lot for the suggestion about evaluation. We can further evaluate the impact of router initialisation on the overall initialisation of a coupler, and can also involve MCT for this test, when revising the manuscript.

Best regards,

Li
* * *

---

## Short Comment (SC4) · 1 May 2020

Dear Editor,

Thanks a lot for your suggestions.

We will add DaRong and the version number to the title when revising the manuscript.

Best regards,

Li
* * *
[Figure]

2020.

---

## Referee Comment (RC3) · VIJAY MAHADEVAN (Referee) · 13 May 2020

**1  Summary**

The manuscript under review presents an improved and scalable routing network generation algorithm for model coupling between component in climate solvers implemented within the C-Coupler2 infrastructure.

The authors present motivations on why the existing routing network generation algorithms do not scale well, and showcase the performance degradation in terms of both time and memory on increasing core counts due to a $O(N^2/K)$ complexity, where $N$

is grid size and $K$ is number of processes. The DaRong algorithm introduces changes to the workflow and corresponding datastructure modifications to maintain the computational cost bounded by $O(Nlog(N)/K)$.

Results demonstrating the superior parallel efficiency of the modified algorithm are also presented in comparison to the global algorithm that involves collective gather/broadcast operations. The figures provided in the manuscript are helpful, and the tables explain the construction of the datastructures needed for the algorithmic workflow. However, the language is confusing in certain sections, and should be rephrased to better improve the overall thesis presented.

In summary, the authors prove that the existing algorithms can be improved through the introduction of an intermediate distribution to eliminate collectives to make this step scalable. However, I fail to see enough conclusive evidence that improvement of just this initialization phase of the solver would lead to a significant impact on the total time to compute the overall coupled climate solution. Additionally, the proposed modifications to the global routing table generation scheme is incremental in nature, and does not aim to minimize communication times between source and destination processes. I do not recommend the publication of the submitted manuscript in the journal of Geoscientific Model Development, as significant new additions are needed to provide better motivations, and results to highlight the overall impact due to the modification in the routing network generation algorithms. Detailed comments are provided below.

**2 General Comments**

The algorithmic modifications proposed in the current manuscript is an incremental update to an existing algorithm, and does not provide a significant enough impact on the overall runtime of the climate solver. It does provide a lower bound on the overall parallel setup cost of the routing network generation, which is used for efficient data-

transfer at runtime during temporal integration of the coupled components. However, since the algorithm does not aim to provide a better partitioning strategy, or make use of architecture layout to minimize communication latency (using say task mapping algorithms), the resulting communication graph between processes still remain the same as the one generated from global routing network algorithm.

More specific comments detailing areas that should be addressed are listed here. I am happy to engage in a conversation if the authors require more clarifications on my comments.

1. First, the authors claim that "existing couplers employ an inefficient and unscalable global implementation for routing network generation that relies on collective communications. That's a main reason why the initialization cost of a coupler increases rapidly when using more processor cores.".

   Can you provide some actual timings from the fully coupled climate solver runs to put the actual setup costs in perspective ? The scalability shown in Fig. (3) still indicate about 3.5s of compute time (since speedup for global routing network case is ≈1) for the 16M grid case on 1600 processes. If this accounts for say over 5% of the actual runtime of the solver, or a non-trivial percentage of total time to simulate a year (or days for high-res) of climate interactions, then 20x improvements in the setup cost could be quite impactful. However, such one time costs get amortized with physics setup costs for high-res runs, in addition to long-term temporal integration of the actual coupled simulation. Hence, I think the manuscript lacks a strong motivation, and provides only an incremental update to avoid the collective algorithms in the coupled simulation invoked during the initial setup phase.

2. Secondly, the global ID based partitioning strategy used in the distributed sort with DaRong to determine the communication pattern is not an innovative concept. There have been several algorithmic ideas based on graph partitioning

strategies used in the parallel Sparse Matrix-Vector (SpMV) linear algebra context [1].

In a simplified sense, without a constraint on the message volume, data locality or latency of communication (such strategies may require repartitioning and/or task mapping), the globally unique ID space can be used in a round-robin type partitioning scheme. For instance, if the source component data are distributed on M processes, and destination on N processes, an implicit decomposition can be determined a-priori based on the global ID numbering that leads to MxN data redistribution. Such an implicit ID decomposition establishes a direct point-to-point communication pattern after which the CLGMT table can be created on both source and destination processes for further send/receive of DoF data at runtime. There may be a need for multiple rounds of rendezvous communication to establish message size for buffer allocation etc, but such an algorithm can eliminate collectives like broadcast and allreduce operations as necessary for better scalability.

**3   Specific Comments**

1. *Line 29-30: "Although most existing couplers achieve scalable data transfer and data interpolation, i.e., the data transfer and data interpolation generally can be faster when using more processor cores, there is almost no evidence of scalable initialization of a coupler."*

   Total cost of a coupled solver includes both the setup/initialization cost and the runtime remap operator application and data-transfer at every time step per coupled component pair/field. Hence, cost of initialization often gets amortized in a climate simulation run. As mentioned previously, please cite or provide data to substantiate such strong claims, preferably with real results using MCT and

C-Coupler2 runs.

2. *Paragraph starting at line 79* is confusing. Please rephrase the sentence better to clearly describe the particular step and its time complexity that leads to the inefficient and non-scalable implementation of routing network generation.

3. *Line 104: "Specifically, we employ a regular intermediate distribution that evenly distributes the CLGMT entries among processes based on the ascending order of the global cell index. Such an intermediate distribution is not only simple, but also enables to easily achieve the rearrangement to the intermediate distribution via a sorting procedure similar to distributed sort. "*

As noted previously, the GSMap and Router infrastructure in MCT already has such options to redistribute data based on Global ID numbering. This is inherently what has been described here as the intermediate distribution of the CLGMT. The primary difference seems to be that GSMap is a $O(P)$ datastructure that grows with core counts, and is accumulated on all process through a gather on root and a subsequent broadcast. The authors of the current paper are trying to avoid this one-time collective operation, which could be an over optimization considering the total runtime of the climate solver.

4. *The paragraph starting at Line 171* can be rewritten in the context of the distributed sort workflow shown in Fig. (2).

5. It will be helpful to explicitly mention the time and memory complexity for each stage in a table format, for both the global and the DaRong algorithm so that the reader can immediately get a sense of the actual improvement.

6. The weak scalability results shown in Table. (7) are not uniform since the grid sizes are doubled, but the core counts do not, going from 250 to 450 and 900 to 1600. Please rerun these calculations with P=[200,400,800,1600] instead.

**4 Technical Corrections**

1. Section (3.2) title: "Rearranging CLGMT entries intra a component model". Please rephrase. Do you mean to say "between component models" ? Same comment applies to Section (3.5) title as well.

2. Line 175, "SPT" should be "SRT" ?

3. Consistently use "Fig." vs "Figure" to reference figures

**References**

Hendrickson, Bruce, Robert Leland, and Steve Plimpton. "An efficient parallel algorithm for matrix-vector multiplication." International Journal of High Speed Computing 7, no. 01 (1995): 73-88.

---

## Short Comment (SC5) · 16 May 2020

Dear Mr. Vijay Mahadevan,

Thanks a lot for your review. The suggestions and comments in the review will further help us to improve the manuscript.

We feel that, there are some misunderstandings arising from the current manuscript and we may not fully understand some of your comments and suggestions. We therefore wish more discussions with you before revising the manuscript.

In the following, we'd like to pose discussions about some review points. Please show

us if some points in the discussions are wrong or incomplete. Thanks a lot.

1. "However, I fail to see enough conclusive evidence that improvement of just this initialization phase of the solver would lead to a significant impact on the total time to compute the overall coupled climate solution." "The algorithmic modifications proposed in the current manuscript is an incremental update to an existing algorithm, and does not provide a significant enough impact on the overall runtime of the climate solver."

RESPONSE: The impact of initialization phase on total time of a model run is relative. Initialization phase will be trivial when simulating a long time (e.g., hundreds of model days or even hundreds of model years), but may be significant for a short simulation (e.g., several model days or even several model hours generally in weather forecasting). It can be required to start to run a model just for only several model hours in data assimilation for weather forecasting. Regarding an operational model, there is generally a time limitation of producing forecasting result (for example, finishing 5-day forecasting in two hours), no matter of the model resolution, and thus developers always have to carefully optimize various software modules especially when the model resolution gets finer.

C-Coupler2 has been used in an operational air-sea coupled model that must finish 7-day forecasting in two hours. This coupled model consists of 3 component models with about 6 million points in the largest model grid, running on over 3000 cores in operational forecasting. C-Coupler2 took about 1200 seconds for initialization at the beginning, and thus we have been asked for accelerating the initialization stage. That's a motivation for this manuscript. We are sorry that we could not include the corresponding results in the manuscript because the codes of this coupled model cannot be open (according to GMD policies, the code used in the manuscript must be available).

Several months ago, a model developer in China asked us whether C-Coupler2 could couple an Ocean model with about 72,000,000 points in the horizontal grid, as he

was being involved in developing such a model. C-Coupler2 cannot handle such coupling because of not only the significant initialization cost but also the huge memory consumption. Similar to some other couplers, C-Coupler2 utilizes the global grid representation, which means that each process keeps all points of each grid. Corresponding to 72,000,000 points, the memory for keeping such a grid in a process will be more than 6GB (given that each point has four vertexes and the data type is double precision). To address this challenge, we started to develop a distributed grid representation a few months ago. We find that it is always required to generate a new distribution of a grid from an existing distribution of the grid. As a grid distribution is essentially a parallel decomposition, the redistribution of grid points can also be handled by the data transfer functionality. Thus, routing network generation will be more frequently used when initializing the coupler, and the impact of its overhead will be more significant. That's another motivation for this manuscript.

We will try to include these motivations when revising the manuscript.

2. "Additionally, the proposed modifications to the global routing table generation scheme is incremental in nature, and does not aim to minimize communication times between source and destination processes."

RESPONSE: The gather/broadcast based global routing table generation can be viewed as a sequential implementation while this manuscript focuses on how to parallelize it for acceleration. Given K processes, the communication times of gather/broadcast is O(K) for each process. Although the implementation proposed in this manuscript retains communication times at O(K), it decreases the message size per process. In other words, it does not change the overall complexity of routing table generation but makes processes cooperatively work together for acceleration. That is a general way how a parallel optimization accelerates a program. So, this manuscript should not be a new algorithm but a distributed implementation. We will modify the title and the corresponding content when revising the manuscript.
3. "However, since the algorithm does not aim to provide a better partitioning strategy, or make use of architecture layout to minimize communication latency (using say task mapping algorithms), the resulting communication graph between processes still remain the same as the one generated from global routing network algorithm."

RESPONSE: As this manuscript can be view as a parallel optimization of the global routing network generation, it should not change the resulting communication graph between processes. The communication graph is generally determined by the parallel decompositions of the source and target component models and the use of architecture layout to lower communication latency can be further determined by the mapping between the process layout and the architecture layout. For a coupler working as a library, it generally can only input the parallel decompositions of models, but cannot change the parallel decompositions, communication graph and process layout.

4. "First, the authors claim that "existing couplers employ an inefficient and unscalableglobalimplementationforroutingnetworkgenerationthatreliesoncollective communications. That's a main reason why the initialization cost of a coupler increases rapidly when using more processor cores."." "Can you provide some actual timings from the fully coupled climate solver runs to put the actual setup costs in perspective ? The scalability shown in Fig. (3) still indicate about 3.5s of compute time (since speedup for global routing network case is ≈1) for the 16M grid case on 1600 processes. If this accounts for say over 5% of the actual runtime of the solver, or a non-trivial percentage of total time to simulate a year (or days for high-res) of climate interactions, then 20x improvements in the setup cost could be quite impactful. However, such one time costs get amortized with physics setup costs for high-res runs, in addition to long-term temporal integration of the actual coupled simulation. Hence, I think the manuscript lacks a strong motivation, and provides only an incremental update to avoid the collective algorithms in the coupled simulation invoked during the initial setup phase. "

RESPONSE: For the motivations of this manuscript, please refer to the response of the first point in this reply. We will add some performance data about the impact of routing

network generation on the total time of coupler initialization.

We use 16M grid for evaluation to show that DaRong can significantly improve routing network generation under different grid size. We know that it is not a real case to run a 16M-grid simulation on only 1600 processes (cores). According to our experiences, a real model with 16M grid can effectively utilizes 40,000 cores (4,00 points per core) or more. We are sorry that we can only use at most 1600 cores per component model in the evaluation. The results in Fig. 3a, Fig. 3b and Table 7 can indicate that the global routing network generation for 16M grid will take much longer than 3.5s when using such as 40,000 cores.

There are always a number of routing networks generated for different data transfers when initializing a coupled model, and generally more routing networks corresponding to more component models in a coupled model (two-way coupling between a pair of component models generally introduces at least two times of routing network generation). Moreover, as stated in the response of the first point in this reply, the distributed grid representation can make routing network generation more frequently used. The parallel read of a remapping weight file can also utilize routing network generation. We have developed a framework for weakly coupled ensemble data assimilation based on C-Coupler2 recently (please refer to its manuscript https://www.geosci-model-dev-discuss.net/gmd-2020-75/), where a specific C-Coupler2 internal component model of the model ensemble will generate routing networks with each ensemble member. This technique will significantly enlarge the number of cores used by a component model and make routing network generation more frequently used. For example, given that there are 20 ensemble members each of which runs on 4,000 processes (cores), the model ensemble runs on 80,000 processes and it will be involved in at least 40 times of routing network generation with the ensemble members.

5. "Secondly, the global ID based partitioning strategy used in the distributed sort with DaRong to determine the communication pattern is not an innovative concept. There have been several algorithmic ideas based on graph partitioning strategies used in

the parallel Sparse Matrix-Vector (SpMV) linear algebra context [1]." "In a simplified sense, without a constraint on the message volume, data locality or latency of communication (such strategies may require repartitioning and/or task mapping), the globally unique ID space can be used in a round-robin type partitioning scheme. For instance, if the source component data are distributed on M processes, and destination on N processes, an implicit decomposition can be determined a-priori based on the global ID numbering that leads to MxN data redistribution. Such an implicit ID decomposition establishes a direct point-to-point communication pattern after which the CLGMT table can be created on both source and destination processes for further send/receive of DoF data at runtime. There may be a need for multiple rounds of rendezvous communication to establish message size for buffer allocation etc, but such an algorithm can eliminate collectives like broadcast and allreduce operations as necessary for better scalability."

RESPONSE: There are a lot of algorithms or optimizations based on the butterfly communication network. It is not an innovative concept indeed. That's why we just give a simple example (Fig. 2) in the manuscript. The key point of this manuscript is how to develop a distributed implementation to accelerate routing network generation. We will correct the misleading title and the related content when revising the manuscript.

The sparse MxN data transfer has been widely used in existing couplers, and this manuscript focuses on how to accelerate the routing network generation for initializing the data transfer. This manuscript might correspond to "There may be a need for multiple rounds of rendezvous communication to establish message size for buffer allocation". We are sorry as we may not fully understand your point here.

Wish your further comments.

Many thanks again,

Li

---

## Author Comment (AC1) · 24 Jun 2020

First, we'd like to say many thanks to the reviewers. The review comments reveal weak points of this manuscript and give us a lot of suggestions for further revision.

Guided by the review comments, we will try to significantly improve the manuscript in the following main aspects:

1. The title will be modified (for example, to "DaRong1.0: a distributed implementation for improving routing network generation in model coupling") and the content will be modified accordingly.

[Figure]

2. The introduction will be rewritten with more motivations, as the discussions in the replies to the reviewers.

3. Some experimental results based on MCT will be added.

4. Incorrect statements, such as related to "collective communication" will be corrected.

Best regards,

Hao Yu, on behalf of all authors.
* * *

---

## Author Response (AR1)

**Part 1: Responses to Moritz Hanke**

We thank Dr. Moritz Hanke for the comments and suggestions. We have modified the manuscript accordingly. In the following, we will reply them one by one.

1. The presented algorithm is basically a rendezvous algorithm, which uses a distributed directory. This was first introduced in [1]. I assume that you did not know about this paper and therefore did not reference it. This algorithm works by distributing data among the processes in a globally known decomposition. This is called "regular intermediate distribution" in the manuscript. Using this intermediate decomposition, accessing data without knowing the original decomposition and without gathering all data on a single process is easily possible. The use of distributed directories in a similar context is mentioned in the following references [2], [3], and [4].

**Response**: Thanks a lot for introducing distributed directory and the related works. We now know that this work as well as our future works deeply benefits from the general idea of distributed directory. The manuscript has been modified accordingly. Please refer to P4L95, P8L249, P9L257.

2. The manuscript describes the router network generation based on two predefined decompositions from two component models as being a fundamental functionality of a coupler. However, this routing table can also be a by-product, in case both components have different grids and the coupler generates interpolation weights online.

**Response**: A routing table can also make help in data interpolation. For example, MCT employs an interpolation decomposition and the corresponding data rearrangement that also follows on a routing table.

3. Line 30: "there is almost no evidence of scalable initialization of a coupler" [...] "(Craig et al., 2017; Liu et al., 2018)". Both papers mentioned in this context contain figures with initialisation cost measurements (see figure 2 and figure 8 respectively). Scaling behaviour in these figures is indeed sub-optimal. However, the cause for this scaling behaviour is not explicitly attributed by either paper to the router generation. Your manuscript indicates that in fact this is the case without providing evidence. By recreating figure 8 from the second paper with your new router generation implementation, you could have confirmed (at least for the C-Coupler) this. In Hanke et al., 2016 (cited by the manuscript) figure 3 (b) shows good scalability of the overall coupler initialisation for up to 3072 processes per component.

**Response**: Our statements here are incorrect. They have been modified. Please refer to P2L47 to P2L49. Moreover, experimental results based on MCT are added. Please refer to P2L49 and Fig. 2.

4. Line 56: "almost all existing couplers use the following 4 steps for generating a routing network" This is a very strong claim, which I would not support (see [2], [3], and [4]).

**Response**: This incorrect statement has been modified. Please refer to P2L59~P3L62.

5. Paragraph 3.2 and figure 2: This is the description of a basic parallel sorting algorithm. A shorter paragraph and a reference to a respective paper would have been enough.

**Response**: In the discussions, we have introduced the differences between the algorithm in Section 3.2 and a basic distributed sorting algorithm. In spite of the differences, we tried to make Section 3.2 as shorter as possible.

6. Line 167-168: "it only utilizes point-to-point communications and does not rely on any collective communication". The parallel sorting algorithm has a complexity of O(log(n)) and a similar communication pattern as the MPI implementations of various collective communication operations[5]. Therefore, you could argue that you actually did use collective communication, which you implemented using point-to-point communication.

**Response**: This incorrect statement has been modified into "it does not rely on any gather/broadcast communication". Please refer to P6L179.

7. Paragraph 4: This paragraph does not describe how these measurements where generated. Did you do a single run, average of multiple runs or average over multiple executions of the algorithm within a single run? It is possible, that especially the first execution of this algorithm produces for some MPI implementations a much higher run time than the following ones.

**Response**: We used average of multiple runs in our evaluation. Please refer to P7L199 in the revised manuscript.

8. Figure 3: I would have preferred to have absolute runtimes instead of speedups for the evaluation of the individual algorithms and speedups only for the direct comparison between

the two.

**Response**: Figure 3 (current Fig. 4) has been modified accordingly. Please refer to P15.

9. Tables 1 to 6: In my opinion, these tables add no significant value to the understanding of the algorithm.

**Response**: Considering some existing couplers such as MCT, OASIS3-MCT and CPL6/CPL7 can also benefit from DiRong1.0, we propose to reserve these tables that can show more details about the implementation.

10. Table 7: I assume the 1600 cores mean, that you used two toy components with 1600 cores each. I could not find an explicit description of this. The core counts are rather odd. You mentioned that your nodes have 24 processors each. Therefore, I would have assume, that the core counts in your tests are multiples of 24.

**Response**: The first column of Table 7 is the number of cores used in each toy component model. We have corrected it and please refer to P22. We used 1600 cores for each component model, because we can use at most 3200 cores on that computer and try to maximize the core number used by models.

11. Intel MPI library (3.2.2) Why did you use such an old version? In my experiments Intel MPI often performed very poorly. Maybe give the most recent OpenMPI a try.

**Response**: We have corrected for this mistake, the correct MPI version used for the evaluations in the manuscript is Intel MPI library (2018 Update 2). Please refer to P7L198 in the revised manuscript.

12. You propose to use a sorting algorithm with complexity $O(\log(n))$ to generate the distributed directory. In some tests, I have seen that some MPI implementations introduce significant delays with such communication pattern, when being used for the first time in a run.

**Response**: We agree that significant delays can happen, while we would think that is the problem of network. Under an unstable network, any kind of MPI communications may suffer from significant delays.

13. Alternatively, each process could compute for all its local points the destination rank. Using alltoall, you can exchange the number of points that need to be sent, in order to get directly from the original to the intermediate decomposition. Afterwards, alltoallv can redistribute the data in a single communication call. Depending on the MPI implementation alltoall can also have a complexity of O(log(n)). However, very little data is exchanged and it can be highly optimized within the MPI. The communication matrix for the alltoallv is probably very sparse. Hence, it can be implemented by the user using point-to-point communication. In my tests, this approach delivers really good results.

**Response**: It is true that each process can easily compute the destination rank for all its local points, and then can easily prepare the parameters of sendcnts as well as sdisls for using MPI_alltoallv. However, each process cannot compute recvcnts as well as rdispls from its local points without extra communications, while recvcnts is necessary for using MPI_alltoallv. So, extra collective communications will be also required for computing recvcnts.

**Part 2: Responses to Vijay Mahadevan**

We thank Dr. Vijay Mahadevan for the comments and suggestions. We have modified the manuscript accordingly. In the following, we will reply them one by one.

1. "First, the authors claim that "existing couplers employ an inefficient and unscalable global implementation for routing network generation that relies on collective communications. That's a main reason why the initialization cost of a coupler increases rapidly when using more processor cores."." "Can you provide some actual timings from the fully coupled climate solver runs to put the actual setup costs in perspective ? The scalability shown in Fig. (3) still indicate about 3.5s of compute time (since speedup for global routing network case is ≈1) for the 16M grid case on 1600 processes. If this accounts for say over 5% of the actual runtime of the solver, or a non-trivial percentage of total time to simulate a year (or days for high-res) of climate interactions, then 20x improvements in the setup cost could be quite impactful. However, such one time costs get amortized with physics setup costs for high-res runs, in addition to long-term temporal integration of the actual coupled simulation. Hence, I think the manuscript lacks a strong motivation, and provides only an incremental update to avoid the collective algorithms in the coupled simulation invoked during the initial setup phase. "

**Response**: This incorrect statement has been modified and please refer to P2L47~P2L49. Moreover, we measured the impact of routing network generation on the total initialization time of MCT in a coupled model (please refer to Figure 2). We did not evaluate the impact of DiRong1.0 on the total time of a model simulation, because this impact can be relative. We discussed about that and introduced more about the motivation in Section 5 (P8L228~P9L258).

2. Secondly, the global ID based partitioning strategy used in the distributed sort with DaRong to determine the communication pattern is not an innovative concept. There have been several algorithmic ideas based on graph partitioning strategies used in the parallel Sparse Matrix-Vector (SpMV) linear algebra context [1]. In a simplified sense, without a constraint on the message volume, data locality or latency of communication (such strategies may require repartitioning and/or task mapping), the globally unique ID space can be used in a round-robin type partitioning scheme. For instance, if the source component data are distributed on M processes, and destination on N processes, an implicit decomposition can be determined a-priori based on the global ID numbering that leads to MxN data redistribution. Such an implicit ID decomposition establishes a direct point-to-point communication pattern after which the CLGMT table can be created on both source and destination processes for further send/receive of DoF data at runtime. There may be a need for multiple rounds of rendezvous communication to establish message size for buffer allocation etc, but such an algorithm can eliminate collectives like broadcast and allreduce operations as necessary for better scalability.

**Response**: The global ID based partitioning strategy can be viewed as a precondition of the MxN data transfer, which is also a precondition of DiRong1.0. We made a slight modification at P4L99 accordingly. The sparse MxN data transfer has been widely used in existing couplers, and this manuscript focuses on how to accelerate the routing network generation for initializing the MxN data transfer. Current implementations of routing network generation rely on gather/scatter communications, while this manuscript might correspond to "There may be a need for multiple rounds of rendezvous communication to establish message size for buffer allocation". There are a lot of algorithms or optimizations based on the butterfly communication network, which has been stated at P5L138 ~ P5L139. It is not an innovative concept indeed. That's why we just give a simple example (Fig. 3) in the manuscript.

3. Line 29-30: "Although most existing couplers achieve scalable data transfer and data interpolation, i.e., the data transfer and data interpolation generally can be faster when using more processor cores, there is almost no evidence of scalable initialization of a coupler." Total cost of a coupled solver includes both the setup/initialization cost and the runtime remap operator application and data-transfer at every time step per coupled component pair/field. Hence, cost of initialization often gets amortized in a climate simulation run. As mentioned previously, please cite or provide data to substantiate such strong claims, preferably with real results using MCT and C-Coupler2 runs.

**Response**: This incorrect statement has been corrected. Please refer to P2L47~P2L49. We have deleted this description and described the scalability of the current coupler more precisely in Section 1 (Line49). Moreover, we measured the impact of routing network generation on the total initialization time of MCT in a coupled model (please refer to Figure 2) and give the related discussions in Section 5 (P8L228~P9L258).

4. Paragraph starting at line 79 is confusing. Please rephrase the sentence better to clearly describe the particular step and its time complexity that leads to the inefficient and non-scalable implementation of routing network generation.

**Response**: The numbers of steps have been added. Please refer to P3L84 and P3L85.

5. Line 104: "Specifically, we employ a regular intermediate distribution that evenly distributes the CLGMT entries among processes based on the ascending order of the global cell index. Such an intermediate distribution is not only simple, but also enables to easily achieve the rearrangement to the intermediate distribution via a sorting procedure similar to distributed sort. " As noted previously, the GSMap and Router infrastructure in MCT already has such options

to redistribute data based on Global ID numbering. This is inherently what has been described here as the intermediate distribution of the CLGMT. The primary difference seems to be that GSMap is a O(P) data structure that grows with core counts, and is accumulated on all process through a gather on root and a subsequent broadcast. The authors of the current paper are trying to avoid this one-time collective operation, which could be an over optimization considering the total runtime of the climate solver.

**Response**: Yes, we are trying to avoid this "one-time collective operation", because it becomes more important in C-Coupler development, which has been discussed in Section 5 (P8L228~P9L258).

6. The paragraph starting at Line 171 can be rewritten in the context of the distributed sort workflow shown in Fig. (2).

**Response**: This paragraph is about the whole implementation of DiRong1.0 (corresponding to the major steps introduced in Section 3.1), while the original Fig. 2 (current Figure 3) is only about the first major step (Section 3.2) and the fourth major step (Section 3.5).

7. It will be helpful to explicitly mention the time and memory complexity for each stage in a table format, for both the global and the DaRong algorithm so that the reader can immediately get a sense of the actual improvement.

**Response**: Thanks a lot for this suggestion. We do not provide a table in the revised manuscript currently, because the paragraph starting from P3L84 states the complexity of each step in the global implantation. We will add a table if it is still required.

8. The weak scalability results shown in Table. (7) are not uniform since the grid sizes are doubled, but the core counts do not, going from 250 to 450 and 900 to 1600. Please rerun these calculations with P=[200,400,800,1600] instead

**Response**: Table 7 has been updated with new results accordingly. Please refer to P22.

9. Section (3.2) title: "Rearranging CLGMT entries intra a component model". Please rephrase. Do you mean to say "between component models" ? Same comment applies to Section (3.5) title as well.

**Response**: We confirm that Section 3.2 and 3.5 are for "intra a component model" but not for

"between component models".

10.  Line 175, "SPT" should be "SRT" ?

**Response**: Thanks a lot. The current P7L187 has been corrected.

Part 3: a marked-up manuscript version

[revised manuscript text omitted]

Formatted Table

---

## Referee Report (RR1)

This version of the paper is a significant improvement.

I still think that the contribution to modelling science is low compared to other papers in this field. The contents may be a great addition to a C-Coupler2 paper. But after discussions with colleagues, I came to the conclusion that by itself it may be interesting to some people.

There are still some minor problems that can be addressed to further improve the paper.

Additionally, I would suggest getting this paper checked by someone with very good English language skills, as I have the feeling that there are still some issues.

L15:

„couplers such as MCT"

I would not call MCT a coupler.

L15:

„inefficient global implementation"

Depending on a number of factors like problem seize, number of processes, and MPI implementation being used, the global implementation may have good performance. Therefore, instead of generally saying that the global implementation is bad, you could for example point out, that your algorithm has significantly better performance characteristics especially for higher processor counts.

L28-29:

"weights that are from an offline file or from online calculation"

What is an offline file?

How about the following?

"weights that read from a file or are calculated online"

L73-87

The analysis of the complexity does not take into account, that MCT allows the use of a compressed global index description, which can significantly reduce memory consumption and time required to detect common grid cells.

Maybe, do the time complexity analysis only for C-Coupler and mention that MCT should be similar but that is also supports compressed indices (I am not exactly sure about the internals of MCT).

L75-76

"corresponding to MCT (as well as 75CPL6/CPL7 and OASIS3-MCT that employ MCT for data transfer)"

Unless you are very familiar with the internal implementation (I am not), you should refrain from making such explicit statements.

---

## Author Response (AR2)

**Part 1: Responses to Hanke Moritz**

We thank Dr. Moritz Hanke for the comments and suggestions. We have modified the manuscript accordingly. In the following, we will reply them one by one.

1. Additionally, I would suggest getting this paper checked by someone with very good English language skills, as I have the feeling that there are still some issues.

Response: Thanks for your suggestion. A native speaker has been invited to improve this manuscript.

2. L15: "couplers such as MCT". I would not call MCT a coupler.

Response: We think that both MCT and C-Coupler can be classified as coupling software. The corresponding statements have been modified in the revised manuscript. Please refer to P1L19 and P3L62.

3. L15: "inefficient global implementation". Depending on a number of factors like problem seize, number of processes, and MPI implementation being used, the global implementation may have good performance. Therefore, instead of generally saying that the global implementation is bad, you could for example point out, that your algorithm has significantly better performance characteristics especially for higher processor counts.

Response: The abstract has been modified accordingly. Please refer to the abstract (P1L20).

4. L28-29: "weights that are from an offline file or from online calculation". What is an offline file? How about the following? "weights that read from a file or are calculated online"

Response: The corresponding statement has been modified accordingly. Please refer to P2L32.

5. L73-87: The analysis of the complexity does not take into account, that MCT allows the use of a compressed global index description, which can significantly reduce memory consumption and time required to detect common grid cells. Maybe, do the time complexity analysis only for C-Coupler and mention that MCT should be similar but that is also supports compressed indices (I am not exactly sure about the internals of MCT). L75-76: "corresponding to MCT (as well as CPL6/CPL7 and OASIS3-MCT that employ MCT for data transfer)" Unless you are very familiar with the internal implementation (I am not), you should refrain from making such explicit statements.

Response: The corresponding context has been modified accordingly. Please refer to P3L76~P3L85.

.

**Part 2: Responses to Anonymous Referee #3**

We thank Anonymous Referee #3 for the comments and suggestions. We have modified the manuscript accordingly. In the following, we will reply them one by one.

1.  I suggest making one more pass with a native English speaker to clean up some rough sections.

Response: Thanks for your suggestion. A native speaker has been invited to improve this manuscript.

2.  It would be helpful if the DiRong lines on figure 4 were clearer. The scaling is completely obscured in 4a and largely obscured in 4b. If the plots cannot be improved, a table might be better. I would also appreciate results on greater than 1600 processor cores, especially for the larger grid sizes. For high resolution cases, the coupler may be run on many more processors than 1600 and it would be good to know whether the performance of the DiRong algorithm rolls over at some processor count as is shown in Figure 4b for the Global results. Results at higher processor counts in at least Figure 4c would improve the paper significantly.

Response: Figure 4 has been replaced by new tables 7 to 10 (P21-P25), where new results corresponding to a grid of 32,000,000 points or 3200 cores for each component model are added. We are sorry of that we can only use a maximum of 6400 cores, while we failed to find a supercomputer in China with more cores available after a lot of efforts in the past four weeks.

Part 3: a marked-up manuscript version

[revised manuscript text omitted]
 routing network generation under using different numbers of cores numbers and athe grid size of 500,000..

[Figure]

(b) The execution time of DiRong1.0 and the global routing network generation usingunder different numbers of cores

 numbers and athe grid size of 4,000,000..

[Figure]

(c) The execution time of DiRong1.0 and the global routing network generation under using different numbers of cores numbers and the a grid size of 16,000,000. .

 **Figure 4. Performance of DiRong1.0 and the comparison with the original global routing network generation (Global) usingunder different numbers of cores numbers and grid sizes. Two toy component models use the same number of**

processor cores in each test case. The comparison of the two algorithms in these figureshere shows that the acceleration effect of DiRong1.0 is more obvious when the number of gridsgrid size and the number of processes is larger,: that isi.e., DiRong1.0 has higher parallel efficiency and better scalability.

**Table 1. The Cell Local-Global Mapping Table (CLGMT) of the parallel decomposition in Fig. 1a**

| Process ID | Cell Local-Global Mapping Table entries |
|---|---|
| 0 | <0,0,0>, <1,0,1>, <8,0,2>, <9,0,3>, <16,0,4>, <17,0,5>, <24,0,6>, <25,0,7> |
| 1 | <2,1,0>, <3,1,1>, <10,1,2>, <11,1,3>, <18,1,4>, <19,1,5>, <26,1,6>, <27,1,7> |
| 2 | <4,2,0>, <5,2,1>, <12,2,2>, <13,2,3>, <20,2,4>, <21,2,5>, <28,2,6>, <29,2,7> |
| 3 | <6,3,0>, <7,3,1>, <14,3,2>, <15,3,3>, <22,3,4>, <23,3,5>, <30,3,6>, <31,3,7> |
| 4 | <32,4,0>, <33,4,1>, <40,4,2>, <41,4,3>, <48,4,4>, <49,4,5>, <56,4,6>, <57,4,7> |
| 5 | <34,5,0>, <35,5,1>, <42,5,2>, <43,5,3>, <50,5,4>, <51,5,5>, <58,5,6>, <59,5,7> |
| 6 | <36,6,0>, <37,6,1>, <44,6,2>, <45,6,3>, <52,6,4>, <53,6,5>, <60,6,6>, <61,6,7> |
| 7 | <38,7,0>, <39,7,1>, <46,7,2>, <47,7,3>, <54,7,4>, <55,7,5>, <62,7,6>, <63,7,7> |

**Table 2. The Cell Local—Global Mapping Table (CLGMT) of the parallel decomposition in Fig. 1b**

| Process ID | Cell Local—Global Mapping Table entries |
|---|---|
| 0 | <0,0,0>, <8,0,1>, <16,0,2>, <24,0,3>, <32,0,4>, <40,0,5>, <48,0,6>, <56,0,7> |
| 1 | <1,1,0>, <9,1,1>, <17,1,2>, <25,1,3>, <33,1,4>, <41,1,5>, <49,1,6>, <57,1,7> |
| 2 | <2,2,0>, <10,2,1>, <18,2,2>, <26,2,3>, <34,2,4>, <42,2,5>, <50,2,6>, <58,2,7> |
| 3 | <3,3,0>, <11,3,1>, <19,3,2>, <27,3,3>, <35,3,4>, <43,3,5>, <51,3,6>, <59,3,7> |
| 4 | <4,4,0>, <12,4,1>, <20,4,2>, <28,4,3>, <36,4,4>, <44,4,5>, <52,4,6>, <60,4,7> |
| 5 | <5,5,0>, <13,5,1>, <21,5,2>, <29,5,3>, <37,5,4>, <45,5,5>, <53,5,6>, <61,5,7> |
| 6 | <6,6,0>, <14,6,1>, <22,6,2>, <30,6,3>, <38,6,4>, <46,6,5>, <54,6,6>, <62,6,7> |
| 7 | <7,7,0>, <15,7,1>, <23,7,2>, <31,7,3>, <39,7,4>, <47,7,5>, <55,7,6>, <63,7,7> |

**Table 3. The distributed CLGMT after rearranging the CLGMT entries in Table 2**

| Process ID | CLGMT entries |
|---|---|
| 0 | <0,0,0>, <1,1,0>, <2,2,0>, <3,3,0>, <4,4,0>, <5,5,0>, <6,6,0>, <7,7,0> |
| 1 | <8,0,1>, <9,1,1>, <10,2,1>, <11,3,1>, <12,4,1>, <13,5,1>, <14,6,1>, <15,7,1> |
| 2 | <16,0,2>, <17,1,2>, <18,2,2>, <19,3,2>, <20,4,2>, <21,5,2>, <22,6,2>, <23,7,2> |
| 3 | <24,0,3>, <25,1,3>, <26,2,3>, <27,3,3>, <28,4,3>, <29,5,3>, <30,6,3>, <31,7,3> |
| 4 | <32,0,4>, <33,1,4>, <34,2,4>, <35,3,4>, <36,4,4>, <37,5,4>, <38,6,4>, <39,7,4> |
| 5 | <40,0,5>, <41,1,5>, <42,2,5>, <43,3,5>, <44,4,5>, <45,5,5>, <46,6,5>, <47,7,5> |
| 6 | <48,0,6>, <49,1,6>, <50,2,6>, <51,3,6>, <52,4,6>, <53,5,6>, <54,6,6>, <55,7,6> |
| 7 | <56,0,7>, <57,1,7>, <58,2,7>, <59,3,7>, <60,4,7>, <61,5,7>, <62,6,7>, <63,7,7> |

385

**Table 4. The Sharing Relationship Table (SRT) calculated from the rearranged distributed CLGMT entries in Fig. 3 and Table 3.**

| Process ID | Sharing Relationship Table entries |
|---|---|
| 0 | <0,0,0,0,0>, <1,0,1,1,0>, <2,1,0,2,0>, <3,1,1,3,0>, <4,2,0,4,0>, <5,2,1,5,0>, <6,3,0,6,0>, <7,3,1,7,0> |
| 1 | <8,0,2,0,1>, <9,0,3,1,1>, <10,1,2,2,1>, <11,1,3,3,1>, <12,2,2,4,1>, <13,2,3,5,1>, <14,3,2,6,1>, <15,3,3,7,1> |
| 2 | <16,0,4,0,2>, <17,0,5,1,2>, <18,1,4,2,2>, <19,1,5,3,2>, <20,2,4,4,2>, <21,2,5,5,2>, <22,3,4,6,2>, <23,3,5,7,2> |
| 3 | <24,0,6,0,3>, <25,0,7,1,3>, <26,1,6,2,3>, <27,1,7,3,3>, <28,2,6,4,3>, <29,2,7,5,3>, <30,3,6,6,3>, <31,3,7,7,3> |
| 4 | <32,4,0,0,4>, <33,4,1,1,4>, <34,5,0,2,4>, <35,5,1,3,4>, <36,6,0,4,4>, <37,6,1,5,4>, <38,7,0,6,4>, <39,7,1,7,4> |
| 5 | <40,4,2,0,5>, <41,4,3,1,5>, <42,5,2,2,5>, <43,5,3,3,5>, <44,6,2,4,5>, <45,6,3,5,5>, <46,7,2,6,5>, <47,7,3,7,5> |
| 6 | <48,4,4,0,6>, <49,4,5,1,6>, <50,5,4,2,6>, <51,5,5,3,6>, <52,6,4,4,6>, <53,6,5,5,6>, <54,7,4,6,6>, <55,7,5,7,6> |
| 7 | <56,4,6,0,7>, <57,4,7,1,7>, <58,5,6,2,7>, <59,5,7,3,7>, <60,6,6,4,7>, <61,6,7,5,7>, <62,7,6,6,7>, <63,7,7,7,7> |

**Table 5. The SRT entries distributed in the *src* component model after rearranging the SRT in Table 4**

| Process ID | Sharing Relationship Table entries |
|---|---|
| 0 | <0,0,0,0,0>, <1,0,1,1,0>, <8,0,2,0,1>, <9,0,3,1,1>, <16,0,4,0,2>, <17,0,5,1,2>, <24,0,6,0,3>, <25,0,7,1,3> |
| 1 | <2,1,0,2,0>, <3,1,1,3,0>, <10,1,2,2,1>, <11,1,3,3,1>, <18,1,4,2,2>, <19,1,5,3,2>, <26,1,6,2,3>, <27,1,7,3,3> |
| 2 | <4,2,0,4,0>, <5,2,1,5,0>, <12,2,2,4,1>, <13,2,3,5,1>, <20,2,4,4,2>, <21,2,5,5,2>, <28,2,6,4,3>, <29,2,7,5,3> |
| 3 | <6,3,0,6,0>, <7,3,1,7,0>, <14,3,2,6,1>, <15,3,3,7,1>, <22,3,4,6,2>, <23,3,5,7,2>, <30,3,6,6,3>, <31,3,7,7,3> |
| 4 | <32,4,0,0,4>, <33,4,1,1,4>, <40,4,2,0,5>, <41,4,3,1,5>, <48,4,4,0,6>, <49,4,5,1,6>, <56,4,6,0,7>, <57,4,7,1,7> |
| 5 | <34,5,0,2,4>, <35,5,1,3,4>, <42,5,2,2,5>, <43,5,3,3,5>, <50,5,4,2,6>, <51,5,5,3,6>, <58,5,6,2,7>, <59,5,7,3,7> |
| 6 | <36,6,0,4,4>, <37,6,1,5,4>, <44,6,2,4,5>, <45,6,3,5,5>, <52,6,4,4,6>, <53,6,5,5,6>, <60,6,6,4,7>, <61,6,7,5,7> |
| 7 | <38,7,0,6,4>, <39,7,1,7,4>, <46,7,2,6,5>, <47,7,3,7,5>, <54,7,4,6,6>, <55,7,5,7,6>, <62,7,6,6,7>, <63,7,7,7,7> |

390

**Table 6. The SRT entries distributed in the *dst* component model after rearranging the SRT in Table 4**

| Process ID | Sharing Relationship Table entries |
|:---:|:---|
| 0 | <0,0,0,0,0>, <8,0,2,0,1>, <16,0,4,0,2>, <24,0,6,0,3>, <32,4,0,0,4>, <40,4,2,0,5>, <48,4,4,0,6>, <56,4,6,0,7> |
| 1 | <1,0,1,1,0>, <9,0,3,1,1>, <17,0,5,1,2>, <25,0,7,1,3>, <33,4,1,1,4>, <41,4,3,1,5>, <49,4,5,1,6>, <57,4,7,1,7> |
| 2 | <2,1,0,2,0>, <10,1,2,2,1>, <18,1,4,2,2>, <26,1,6,2,3>, <34,5,0,2,4>, <42,5,2,2,5>, <50,5,4,2,6>, <58,5,6,2,7> |
| 3 | <3,1,1,3,0>, <11,1,3,3,1>, <19,1,5,3,2>, <27,1,7,3,3>, <35,5,1,3,4>, <43,5,3,3,5>, <51,5,5,3,6>, <59,5,7,3,7> |
| 4 | <4,2,0,4,0>, <12,2,2,4,1>, <20,2,4,4,2>, <28,2,6,4,3>, <36,6,0,4,4>, <44,6,2,4,5>, <52,6,4,4,6>, <60,6,6,4,7> |
| 5 | <5,2,1,5,0>, <13,2,3,5,1>, <21,2,5,5,2>, <29,2,7,5,3>, <37,6,1,5,4>, <45,6,3,5,5>, <53,6,5,5,6>, <61,6,7,5,7> |
| 6 | <6,3,0,6,0>, <14,3,2,6,1>, <22,3,4,6,2>, <30,3,6,6,3>, <38,7,0,6,4>, <46,7,2,6,5>, <54,7,4,6,6>, <62,7,6,6,7> |
| 7 | <7,3,1,7,0>, <15,3,3,7,1>, <23,3,5,7,2>, <31,3,7,7,3>, <39,7,1,7,4>, <47,7,3,7,5>, <55,7,5,7,6>, <63,7,7,7,7> |

**Table 7. Performance of DiRong1.0 and the comparison with the original global routing network generation (Global) using different numbers of cores numbers and the grid size of 500,000.**

| Core number of each toy component model | DiRong1.0 | | Global | | Global/DiRong1.0 |
|---|---|---|---|---|---|
| | Time (s) | Speedup | Time (s) | Speedup | |
| 60 | 0.031 | 1.000 | 0.129 | 1.000 | 4.110 |
| 120 | 0.040 | 0.774 | 0.278 | 0.462 | 6.888 |
| 240 | 0.047 | 0.671 | 0.243 | 0.530 | 5.205 |
| 480 | 0.029 | 1.076 | 0.478 | 0.269 | 16.461 |
| 960 | 0.033 | 0.943 | 1.169 | 0.110 | 35.224 |
| 1600 | 0.034 | 0.912 | 1.737 | 0.074 | 50.641 |
| 3200 | 0.036 | 0.862 | 2.573 | 0.050 | 70.900 |

395

**Table 8. Performance of DiRong1.0 and the comparison with the original global routing network generation (Global) using different numbers of cores numbers and the grid size of 4,000,000.**

| Core number of each toy component model | DiRong1.0 | | Global | | Global/DiRong1.0 |
|---|---|---|---|---|---|
| | Time (s) | Speedup | Time (s) | Speedup | |
| 60 | 0.161 | 1.000 | 0.863 | 1.000 | 5.349 |
| 120 | 0.117 | 1.375 | 0.517 | 1.668 | 4.409 |
| 240 | 0.081 | 1.990 | 0.437 | 1.974 | 5.391 |
| 480 | 0.060 | 2.669 | 0.649 | 1.329 | 10.737 |
| 960 | 0.051 | 3.184 | 1.308 | 0.660 | 25.811 |
| 1600 | 0.045 | 3.548 | 1.949 | 0.443 | 42.858 |
| 3200 | 0.039 | 4.098 | 2.623 | 0.329 | 66.598 |

400

**Table 9. Performance of DiRong1.0 and the comparison with the original global routing network generation (Global) using different numbers of cores numbers and the grid size of 16,000,000.**

| Core number of each toy component model | DiRong1.0 | | Global | | Global/DiRong1.0 |
|---|---|---|---|---|---|
| | Time (s) | Speedup | Time (s) | Speedup | |
| 60 | 0.702 | 1.000 | 3.437 | 1.000 | 4.899 |
| 120 | 0.447 | 1.571 | 2.351 | 1.462 | 5.263 |
| 240 | 0.276 | 2.547 | 2.363 | 1.455 | 8.575 |
| 480 | 0.169 | 4.163 | 2.529 | 1.359 | 15.006 |
| 960 | 0.109 | 6.429 | 3.135 | 1.097 | 28.721 |
| 1600 | 0.106 | 6.628 | 3.065 | 1.121 | 28.956 |
| 3200 | 0.098 | 7.133 | 3.242 | 1.060 | 32.960 |

405

**Table 10. Performance of DiRong1.0 and the comparison with the original global routing network generation (Global) using different numbers of cores numbers and the grid size of 32,000,000.**

| Core number of each toy component model | DiRong1.0 | | Global | | Global/DiRong1.0 |
|---|---|---|---|---|---|
| | Time (s) | Speedup | Time (s) | Speedup | |
| 60 | 1.438 | 1.000 | 6.878 | 1.000 | 4.782 |
| 120 | 0.960 | 1.499 | 4.206 | 1.635 | 4.383 |
| 240 | 0.554 | 2.597 | 4.739 | 1.451 | 8.557 |
| 480 | 0.340 | 4.234 | 5.083 | 1.353 | 14.964 |
| 960 | 0.199 | 7.222 | 6.098 | 1.128 | 30.616 |
| 1600 | 0.176 | 8.182 | 5.758 | 1.195 | 32.756 |
| 3200 | 0.165 | 8.704 | 5.500 | 1.251 | 33.286 |

**Table 711. Performance of DiRong1.0 and the comparison with the original global routing network generation (Global) when concurrently increasing the grid size and number of cores number.**

| Core number of each toy component model | Grid size | Execution time (s) of DiRong1.0 | Execution time (s) of Global | Global/ DiRong1.0 |
|---|---|---|---|---|
| 250 | 500,000 | 0.032 | 0.262 | 8.19 |
| 450 | 1,000,000 | 0.034 | 0.492 | 14.47 |
| 900 | 2,000,000 | 0.041 | 1.158 | 28.24 |
| 1600 | 4,000,000 | 0.045 | 1.949 | 43.31 |
| 3200 | 8,000,000 | 0.063 | 2.850 | 45.24 |

410

---

## Author Response (AR3)

**Part 1: Responses to Topical Editor**

Dear Editor,

Thanks a lot for your comments and suggestions. We have modified the manuscript accordingly. In the following, we will reply one by one.

1. Figure 1: I don't understand the correspondence between c) and a) and b). Figure 1 a) represents a component with 64 grid points in total decomposed over 8 processes with the first process holding (in the global index space) grid points 0,1,8,9,16,17,24,25, the second process holding grid points 2,3,10,11,18,19,26,27, and so on. Figure c) on the left represents (for me) a component with 16 grids points in total decomposed over 8 processes with the first process S0 holding (in the global index space) grid points 0 and 1, the second process S1 holding grid points 2 and 3, the third process S2 holding grid points 4 and 5, and so on. So a) does not correspond to the left of figure c, and b) does not correspond to the right of figure c. Why do you associate them in Figure 1? Please rectify or explain.

Response: The title of Figure 1 has been modified accordingly. Please refer to P12.

2. p.4, l.111-115: I really don't understand the sentence: "Existing global implementations depend on global CLGMTs because the distribution of the CLGMT entries is determined by a model, and thus a coupler generally has to view any distribution as random." , could you clarify?

Response: We find that some content of this sentence seems useless. We modified the paragraph from P4L108 to P4L112.

3. Figure 2 captions: What does "and a regular 1-D parallel decomposition is designed for the data interpolation." mean? Please clarify.

Response: We find that this statement is useless and have removed it. Please refer to P13.

4. p.3 l.81-85: please consider changing this sentence for : "... similar complexities to C-Coupler, even if a compressed global index description is, in the case of regular parallel decompositions, used to reduce the memory and time required to detect common grid cells (the compressed description may not work for irregular, such as round-robin, parallel decompositions)

Response: The sentence has been modified according to your suggestions. Please refer to P3L81~P3L85.

5. p.3, l.76: you assume that both src and dst components use the same number of processes; this is OK but I would then change "Given that ..." by "Assuming that ..."

Response: We have modified the manuscript accordingly. Please refer to P3L76

6. In the Abstract, l.19, maybe you should write C-Coupler1 to contrast with C-Coupler2 into which the new DiRong scheme has been implemented, as stated later in the paragraph.

Response: The existing versions of C-Coupler use the global implementation. The abstract is modified accordingly. Please refer to P1L19.

7. Figure 3 captions: I think the comma should be removed between "entries" and "larger"

Response: The manuscript has been modified accordingly. Please refer to Figure 3 in P14.

`

Part 2: a marked-up manuscript version

[revised manuscript text omitted]

---

## Author Response (AR4)

**Part 1: Responses to Topical Editor**

Dear Editor,

Thanks a lot for your new suggestions. We have modified the manuscript accordingly. Please refer to

P4L109 and P3L83, respectively. Moreover, we have updated our code availability section.

    With best regards,

Hao

Part 2: a marked-up manuscript version

[revised manuscript text omitted]